



# Machine learning as an inversion algorithm for aerosol profile and property retrieval from multi-axis differential absorption spectroscopy measurements: a feasibility study

Yun Dong[1], Elena Spinei[1], Anuj Karpatne[2]

[1]Department of Electrical and Computer Engineering, Virginia Tech, Blacksburg, VA 24060, USA
[2]Department of Computer Science, Virginia Tech, Blacksburg, VA 24060, USA

*Correspondence to*: Elena Spinei (eslind@vt.edu)

**Abstract.** In this study, we explore a new approach based on machine learning (ML) for deriving aerosol extinction coefficient profiles, single scattering albedo and asymmetry parameter at 360 nm from a single MAX-DOAS sky scan. Our method relies on a multi-output sequence-to-sequence model combining Convolutional Neural Networks (CNN) for feature extraction and Long Short-Term Memory networks (LSTM) for profile prediction. The model was trained and evaluated using data simulated by VLIDORT v2.7, which contains 1459200 unique mappings. 75% randomly selected simulations were used for training and the remaining 25% for validation. The overall error of estimated aerosol properties for (1) total AOD is -1.4 ± 10.1 %, (2) for single scattering albedo is 0.1 ± 3.6 %; and (3) asymmetry factor is -0.1 ± 2.1 %. The resulting model is capable of retrieving aerosol extinction coefficient profiles with degrading accuracy as a function of height. The uncertainty due to the randomness in ML training is also discussed.

## 1. Introduction

Aerosols play an important role in the Earth-atmosphere system by modifying the global energy balance, participating in cloud formation and atmospheric chemistry, and fertilizing land and ocean. Aerosols are widely spread in the troposphere and are emitted by anthropogenic and natural processes (primary aerosols), and are formed by gas-to-particle conversion mechanisms (secondary aerosols). Aerosols are removed from the atmosphere by dry (gravitational settling and turbulent) deposition and wet deposition, and have relatively short lifetimes ranging from a few minutes to a few weeks (Haywood and Boucher, 2000). The aerosol classification depends on the aerosol source, composition, size and number distribution, aging processes, and optical and physical properties.

The spatial and temporal distribution of aerosols in the lower troposphere is highly variable and is greatly affected by the proximity to the sources, type of aerosols, meteorological conditions, and photochemical processes. Horizontal and vertical heterogeneity of the aerosol distribution, their properties and processes pose a serious challenge for modeling aerosol induced radiative forcing and is an important source of uncertainties in the climate modeling results (Intergovernmental Panel on Climate Change, 2014).

Macroscopic aerosol optical properties required for modeling aerosol radiative forcing include single scattering albedo, scattering phase function, and aerosol optical thickness (AOD), (Dubovik et al., 2002).





These parameters depend on aerosol chemical composition, aerosol mixing, particle shape and size distribution, and particle orientation.

Single scattering albedo, $\omega(\lambda)$, is defined as the ratio of scattering optical depth ($\tau_{scattering}$) to the total optical depth ($\tau_{scattering} + \tau_{absorption}$) at wavelength $\lambda$ (Eq. (1)):

$$\omega(\lambda) = \frac{\tau_{scattering}}{\tau_{scattering} + \tau_{absorption}} \, , \tag{1}$$

The magnitude of $\omega(\lambda)$ determines whether the aerosols have a cooling or warming effect depending on the underlying surface albedo. Since $\omega(\lambda)$ mainly depends on the aerosol composition (complex part of the refractive index) and size, it is difficult to characterize for aerosol mixtures, especially of the anthropogenic origin.

Scattering phase function describes the angular intensity distribution of electromagnetic radiation scattered by the aerosol. It depends on the aerosol size compared to the incident electromagnetic radiation wavelength ($\lambda$), aerosol particle shape, and composition (relative refractive index $m$ at $\lambda$). In the Lorenz-Mie formalism, applied in this study, wavelength-aerosol size dependence is expressed by the size parameter ($\alpha$) as the ratio of the spherical particle circumference to the wavelength (Seinfeld and Pandis, 2016).

The scattering phase function, $P(\theta,\alpha,m)$, at a scattering angle $\theta$ for spheres is calculated by normalizing the scattered intensity into angle $\theta$ by the intensity integrated over all scattering directions. The dominating scattering direction is described by the asymmetry factor ($g$), which is defined as the phase function weighted cosine of the scattering angles integrated from 0° (forward direction) to 180° (backward direction):

$$g(\alpha, m) = \frac{1}{2} \int_0^\pi \cos(\theta) \cdot P(\theta, \alpha, m) \cdot \sin(\theta) d\theta \, , \tag{2}$$

The asymmetry factor ranges from -1 (backscattering) to +1 (forward scattering). Henyey and Greenstein (1941) proposed a simplified "fitting" technique to calculate P($\theta$) using solely the asymmetry factor:

$$P_{HG}(\cos\theta) = \frac{1 - g^2}{(1 + g^2 - 2 \cdot g \cdot \cos(\theta))^{\frac{3}{2}}} \, , \tag{3}$$

Several methods used to solve the radiative transfer equation in the atmosphere (e.g. $\delta$-M, discrete ordinate, and Monte Carlo) require scattering phase function expansion into a finite series of Legendre polynomials (PL($\cos\theta$)) to account for the dependence of the radiation field on azimuth (Spurr, 2008). Lorenz-Mie type codes output the Legendre expansion coefficients. The expansion of the Henyey-Greenstein phase function into Legendre polynomials ($P_L$) is given by a simple relationship shown in Eq. (4), where $(2L+1)g^L$ is its Legendre moments (expansion coefficients).

$$P_{HG}(\cos\theta) = \sum(2L + 1) \cdot g^L \cdot P_L(\cos\theta), \tag{4}$$

## 2. Multi-Axis Differential Optical Absorption (MAX-DOAS) technique



MAX-DOAS technique has been widely used to derive vertical aerosol extinction coefficient profiles in the lower troposphere. This is typically done from ground-based measurements of oxygen collision complex ($O_2O_2$) absorption (for a detailed list of references see Table 1 in Wagner et al., (2018)). Since the oxygen volume mixing ratio ($\chi_{O2} = 0.209$) is considered constant, the $O_2O_2$ abundance depends only on the total number of air molecules (pressure, temperature and to a small degree humidity) and can be easily calculated.

More than 93% of $O_2O_2$ is located below 10 km (scale height ~ 4 km). Any deviation in measured $O_2O_2$ absorption from this molecular (Rayleigh) scattering case is only due to the change in the photon path through the $O_2O_2$ layer. Aerosols and clouds are the main causes of such photon path modification for ground-based measurements. $O_2O_2$ has several absorption bands in the ultraviolet (UV) and visible (VIS) parts of the electromagnetic spectrum (band peaks at 343, 360, 380, 477, 577, 630 nm (Thalman and Volkamer, 2013).

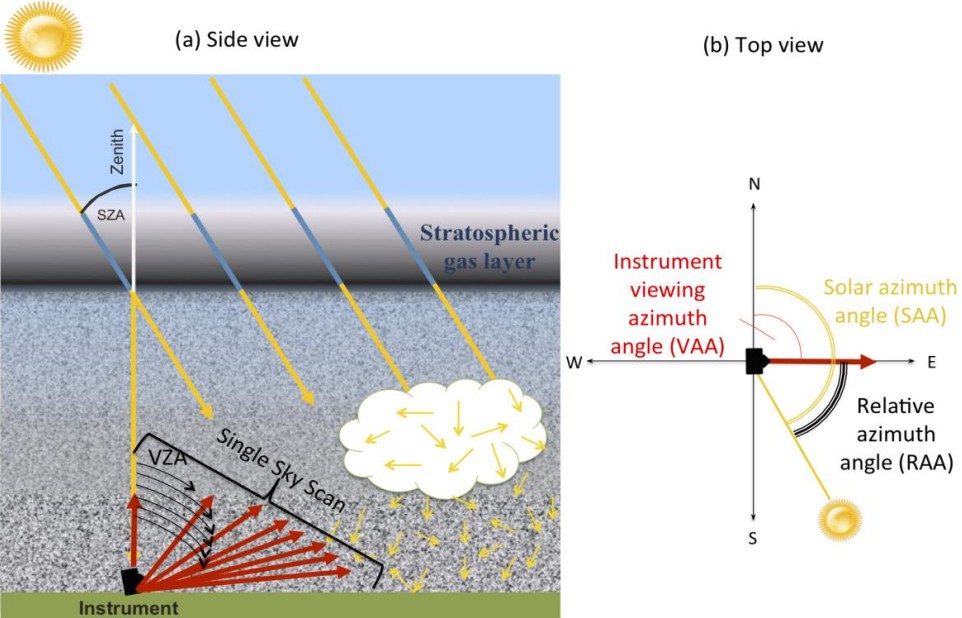

**Figure 1. Demonstration of the MAX-DOAS principle: (a) side view and (b) top view. Simplified photon paths through the atmosphere are shown in yellow. A single sky scans sequence for profile retrieval consists of multiple viewing zenith angles (VZA) in a specific direction (viewing azimuth angle, VAA) at a specific solar zenith angle (SZA) and is shown in red.**

The MAX-DOAS technique consists of measuring sky-scattered UV-VIS solar spectra at multiple, primarily, low elevation angles (Fig. 1). MAX-DOAS shows a large sensitivity to the tropospheric gases due to increased photon path length through the lower troposphere (Platt and Stutz, 2008). To eliminate the contribution from the upper atmosphere solar spectra measured at low elevation angles are divided by the reference spectrum collected from the zenith direction. DOAS technique has the advantage of not needing an

absolute radiometric calibration.

The first step of the DOAS retrieval is a spectral evaluation to calculate the differential slant column density ($\Delta SCD_{measured} = SCD - SCD_{reference}$) of $O_2O_2$. This step is accomplished through the simultaneous non-linear





least-squares fitting of the absorption by species *i*, low-order polynomial function ($P_{LO}$) and offset to the difference between the logarithms of the attenuated (*I*) and reference (*I_{reference}*) spectra. $P_{LO}$ estimates combined attenuation due to molecular scattering and aerosol total extinction (scattering and absorption). Offset term approximates instrumental stray light and residual dark current.

$$ln\left(I_{reference}(\lambda)\right) - ln\left(I(\lambda) - offset(\lambda)\right) = \left(\sum_s \sigma_i(\lambda) \cdot \Delta SCD_i\right) + P_{LO},$$ (5)

The second step of the MAX-DOAS analysis is the conversion of a single sky scan (multiple viewing angles) $\Delta SCD(O_2O_2)$ into a vertical aerosol extinction coefficient profile. The physical relationship between the measured $\Delta SCD$ and the desired aerosol extinction coefficient profile and aerosol properties is complex, and, in general, can be expressed mathematically by equation (6) (Rodgers, 2004):

$$\boldsymbol{y} = f(\boldsymbol{x}, \boldsymbol{b}) + \boldsymbol{\varepsilon},$$ (6)

Where, the measured quantities (measurement vector $\boldsymbol{y}$) are described by a forward model $f(\boldsymbol{x}, \boldsymbol{b})$ and the measurement error vector ($\boldsymbol{\varepsilon}$). The forward model, $f(\boldsymbol{x}, \boldsymbol{b})$, is a model that estimates physical processes that relate the measured parameter ($\boldsymbol{y}$), the unknown quantity to be retrieved (state vector ($\boldsymbol{x}$)), and forward model parameters ($\boldsymbol{b}$) that are considered approximately known. Under most conditions, there are more unknowns than measurements, and as a result equation (6) does not have a unique solution.

The inversion of equation (6) is often done in the framework of Bayes' theorem, which allows for the assignment of probability density functions to all possible states given measurements and prior knowledge of the state. However, in reality, we are not interested in all possible solutions, but rather a single, the most "probable" solution with its error estimation. Equation (7) shows a Transfer Function that defines an estimated solution ($\hat{\boldsymbol{x}}$) as a function of the measurement system and retrieval method (Rodgers, 2004):

$$\hat{\boldsymbol{x}} = R\left(f(\boldsymbol{x}, \boldsymbol{b}) + \boldsymbol{\varepsilon}, \hat{\boldsymbol{b}}, \boldsymbol{x}_a, \boldsymbol{c}\right),$$ (7)

where $R$ is a retrieval method, $f(\boldsymbol{x}, \boldsymbol{b})$ is a forward function with the true state ($\boldsymbol{x}$) and true parameters ($\boldsymbol{b}$), $\hat{\boldsymbol{b}}$ is the estimated forward model parameter vector, $\boldsymbol{x}_a$ is the a priori estimate of state vector ($\boldsymbol{x}$), and $\boldsymbol{c}$ is a retrieval method parameter vector (e.g. convergence criteria). For nonlinear problems the solution to equation (7) cannot be found explicitly, and iterative numerical methods are required. A maximum a posteriori (MAP) approach has been widely applied to moderately nonlinear problems with Gaussian distribution of both measurement errors and a priori state errors. A priori information about the state vector distribution before the measurements are made is used to constrain the solution of the ill-posed problems (Rodgers, 2004). It is essential to use the best estimate of the state available since in the MAP approach the retrieved state is proportional to the weighted mean of the actual state and the a priori state. In addition, an appropriate covariance matrix for the a priori state vector has to be constructed. This a priori information for aerosol vertical extinction coefficient profiles, however, is rarely available.

In addition to the optimal estimation method (OEM), briefly described above, parameterized (Beirle et al., 2019; Vlemmix et al., 2015) and analytic (Spinei et al 2019, in preparation) inversion algorithms were developed. Frieß et al., (2019) provide a detailed intercomparison of currently available state-of-the-art



inversion algorithms for MAX-DOAS measurements. None of the algorithms perform perfectly and none of them estimate asymmetry factor or single scattering albedo in addition to aerosol extinction coefficient profiles. Most of the current algorithms take between 3 to 216 seconds to process a single MAX-DOAS sky scan (Frieß et al., 2019) mainly due to the iterative inversion step. They also require external information about the atmosphere (e.g. temperature and pressure profiles, aerosol single scattering albedo and asymmetry factor) and a priori information that does not typically exist. With an increasing number of MAX-DOAS 2-D instruments worldwide capable of sunrise to sunset measurements (e.g. Pandonia Global Network) fast methods are needed that can harvest full information from the MAX-DOAS hyperspectral measurements.

This study describes and evaluates a fast novel machine learning (ML) approach for retrieving aerosol extinction coefficient profiles, asymmetry factor and single scattering albedo at 360 nm from $\Delta SCD(O_2O_2)$ observations within a single MAX-DOAS sky scan. The basic idea of our approach is: (1) develop an "inverse model" by one-time offline training of a supervised ML algorithm on simulated MAX-DOAS data and corresponding atmospheric aerosol conditions, and (2) use the relationships derived in the first step to estimate the aerosol extinction profile, asymmetry factor, and single scattering albedo from the MAX-DOAS $\Delta SCD(O_2O_2)$ measurements. We specifically leverage recent advances in ML, e.g., deep learning methods, to automatically extract the inverse mapping from the observations ($y$) to the state vectors ($x$), using a collection of ($x, y$) pairs available for training. Different machine learning algorithms were successfully used in remote sensing applications (Schulz et al., 2018, Schilling et al., 2018, Efremenko et al., 2017; Hedelt et al., 2019).

The rest of the paper is organized in the following sections. Section 3 provides an overview of the new retrieval algorithm. Section 4 focuses on training data generation using the radiative transfer model (VLIDORT). Section 5 details ML implementation. Section 6 provides an extensive comparison of ML predicted versus "true" macroscopic aerosol properties outside the training dataset. Section 7 summarizes the findings.

## 3. Overview of the Methodology

Our approach consists of two stages: (1) a one-time training stage that results in an inverse ML model $R(\widehat{\Theta})$ with appropriate architecture and parameters $\widehat{\Theta}$ ; and (2) an inversion stage, where the trained ML model $R(\widehat{\Theta})$ is applied to MAX-DOAS measurements to retrieve aerosol properties. Figure 2 provides a schematic overview of both stages.

The offline training stage comprises of two key steps. First, a training set containing simulated measurements $\{y_i | i = 1,2, ... , N\}$ is generated by a forward model (VLIDORTv2.7) given atmospheric states $\{x_i | i = 1,2, ... , N\}$. The model describes atmospheric radiative transfer processes connecting the atmospheric states and the measurements. Second, both the atmospheric states and the simulated measurements are fed into the ML model for learning the inverse mapping from the measurement space to the state space. This is based on solving an optimization problem that minimizes the mean squared error (MSE) between the retrieved values





($\{\hat{x}_i | i = 1, 2, \ldots, N\}$) and the true values ($\{x_i | i = 1, 2, \ldots, N\}$). We specifically chose artificial neural network (ANN) models to learn the inverse mapping from $y$ to $x$. By iteratively adjusting the parameters of the ANN

model using gradient descent (backpropagation) algorithms (Johansson et al., 1991), we are able to arrive at ANN model parameters $\hat{\Theta}$ that provide a local optimum performance in terms of MSE on the training data. The result of the training stage is an inverse model $R(\hat{\Theta})$ whose architecture and parameters are saved in an HDF5 file (1.3 MB). The trained model $R(\hat{\Theta})$ is an inversion operator that transforms measurements vector $y$ into the state vector $\hat{x}$ through a set of simple linear and nonlinear operations. The inverse model provides

a convenient and fast way for retrieval of aerosol properties from $\Delta SCD(O_2O_2)$ measurements during the inversion stage. It takes ~0.15 ms for the retrieval of the studied aerosol properties from a single MAX-DOAS sky scan $\Delta SCD(O_2O_2)$ on a single CPU core.

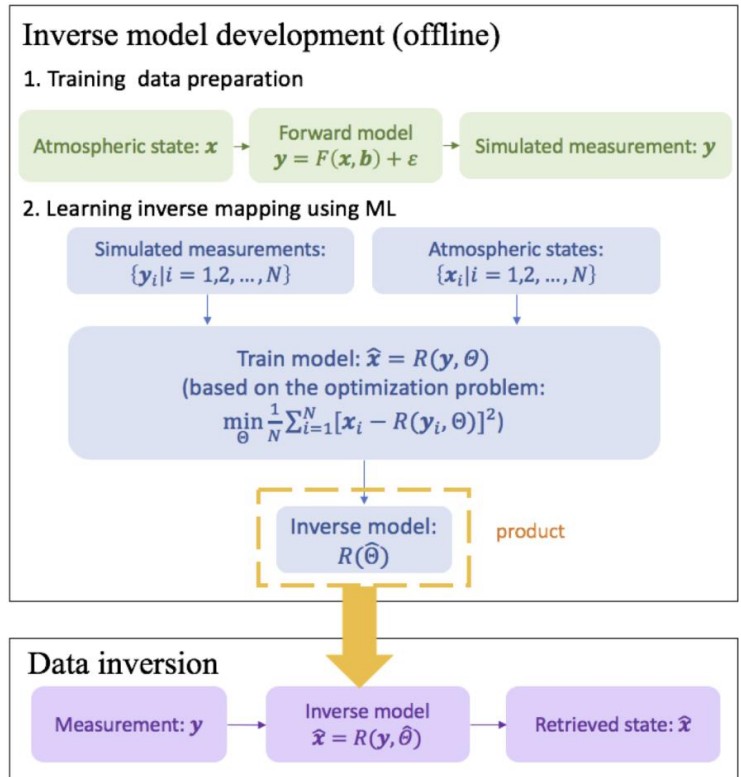

**Figure 2. Schematics of the machine learning inversion algorithm.**

**4. Training data preparation**

The success of any ML model depends on the quality of the training data. Since there is no reliable dataset that combines simultaneous MAX-DOAS measurements and observations of aerosol macrophysical properties and vertical extinction coefficient profiles at 360 nm we use a radiative transfer model to simulate


MAX-DOAS measurements. In this study, we train our ML model on air mass factors (AMF) calculated
from the simulated solar radiances at the bottom of the atmosphere.

AMF represents a ratio between the true average path that photons take through a gas layer before detection
by a MAX-DOAS instrument and the vertical path. Since $O_2O_2$ absorption in the reference (zenith scattered)
spectrum is not precisely known, a differential AMF at a specific wavelength λ and observations geometry μ
(relative azimuth angle, solar zenith angle, and viewing zenith angle), is determined as:

$$\Delta AMF(O_2O_2, \lambda, \mu) = \frac{\Delta SCD_{measured}(O_2O_2, \lambda, \mu)}{VCD(O_2O_2)_{calculated}} = \frac{\ln\left(I_{reference}(\lambda, \mu_o)\right) - \ln(I(\lambda, \mu))}{VCD(O_2O_2)_{calculated} \cdot \sigma(O_2O_2, \lambda)},$$ (8)

Where vertical column density of $O_2O_2$ (VCD) is estimated as the squared oxygen number density integrated
from the surface to the top of the atmosphere; and σ(λ) is the molecular absorption cross-section of $O_2O_2$.

In the absence of aerosols and clouds only air molecules (mainly oxygen and nitrogen) scatter solar photons
in the Earth's atmosphere. This molecular only (Rayleigh) scattering process is considered to be well
understood (Bodhaine et al., 1999) and $\Delta AMF^{Rayleigh}$ can be calculated from the simulated intensities. In the
presence of aerosols, dust and clouds not only air molecules but also particles and cloud droplets scatter solar
photons. This type of scattering can be generally described by the T-matrix theory. In this study we consider
only spherical aerosols (Lorenz-Mie theory), whose scattering phase function is approximate according to
the Henyey-Greenstein approach using the asymmetry factor *g*. $\Delta AMF^{aerosol+Rayleigh}$ are determined from
simulated downwelling radiances for atmosphere with different aerosol types and their extinction coefficient
profiles. The change in AMF due to aerosol presence can be described by $\Delta AMF^{aerosol}$:

$$\Delta AMF^{aerosol} = \Delta AMF^{Rayleigh} - \Delta AMF^{aerosol+Rayleigh},$$ (9)

*ΔAMF^aerosol* for $O_2O_2$ at 360 nm for different observation geometries and scattering conditions is used for ML
training in this feasibility study. A single MAX-DOAS measurement considered here is *ΔAMF^aerosol* set from
the full sky scan at a single solar zenith angle, single relative azimuth angles, and the following viewing
zenith angles: 0, 40, 50, 60, 65, 70, 75, 80, 81, 82, 83, 84, 85, 86, 87, 88, 89°. To ensure that the training
dataset contains all observation geometries feasible for MAX-DOAS sky scans we have included the
following:

(1) Relative azimuth angles: 0, 10, 20, 30, 40, 50, 60, 70, 80, 90, 100, 110, 120, 130, 140, 150, 160, 170,
   180°, and

(2) Solar zenith angles: 0, 10, 20, 30, 40, 50, 60, 65, 70, 75, 80, 85°.

Solar radiances at the bottom of the atmosphere were simulated using VLIDORT v.2.7 (Spurr, 2008).
VLIDORT is a discrete-ordinate radiative transfer model that has been successfully applied to simulate
radiances and weighting functions for forward models in optimal estimation inversion (Clémer et al., 2010)
and machine learning algorithms (Efremenko et al., 2017, Hedelt et al., 2019). VLIDORT code applies
pseudo-spherical approximation to direct solar beam attenuation in a curved atmosphere. All scattering
processes are estimated using the plane-parallel approximation in a stratified atmosphere. Precise single
scattering computation is performed using Nakajima/Tanaka ansatz and delta-M scaling. VLIDORT v.2.7


calculates analytically derived Jacobians (radiance weighting functions) with respect to any profile/column/surface variables. VLIDORT computes elastic scattering by molecules to all orders (Spurr, 2008).

**Table 1. Radiative transfer model settings**

| General Model Settings | Physical and Observation Geometry Inputs |
|---|---|
| NO Refraction correction;<br><br>Scalar calculations;<br><br>Only elastic scattering;<br><br>Aerosol scattering phase function estimation using Henyey-Greenstein approximation from the asymmetry factor (g). | **Observation Geometry:**<br>Viewing zenith angle scan: 0, 40, 50, 60, 65, 70, 75, 80, 81, 82, 83, 84, 85, 86, 87, 88, 89°;<br>Relative azimuth angles: 0, 10, 20, 30, 40, 50, 60, 70, 80, 90, 100, 110, 120, 130, 140, 150, 160, 170, 180°<br>Solar Zenith angles: 0, 10, 20, 30, 40, 50, 60, 65, 70, 75, 80, 85, 86, 87, 88, 89°<br><br>**Wavelength:** 360 nm;<br><br>**Vertical grid (67 layers):**<br>100 m up to 4 km, 500 m from 4 to 8 km, 1 km from 8 to 12km, 2 km from 12 to 30km, 5 km from 30 to 60 km<br><br>**Atmospheric air density:**<br>Pressure [hPa]: US1976 standard atmosphere<br>Temperature [K]: US1976 standard atmosphere<br><br>**Gas volume mixing ratio profiles:**<br>$O_3$ profile: climatology over Cabauw in September<br>$O_3$ molecular absorption cross-section: Daumont<br>$O_2O_2$ profile: from temperature and pressure<br>$O_2O_2$ molecular absorption cross-section: Thalman and Volkamer (2011)<br><br>**Aerosol properties:**<br>Single scattering albedo: 0.775, 0.825, 0.875, 0.925, 0.975<br>Henyey-Greenstein asymmetry factor: 0.675, 0.725, 0.775, 0.825<br><br>**Aerosol extinction coefficient profiles [1/km] as a function of altitude;**<br>Exponential function at the surface combined with "sliding" Gaussian function above;<br>Total AOD: 0, 0.15, 0.3, 0.45, 0.6, 0.75;<br>Gaussian profile center height: 0.5, 1, 1.5, 2 km;<br>Gaussian width: 0.1, 0.2, 0.3, 0.5 km;<br>Partitioning between exponential and Gaussian attributed AOD: 0.3, 0.6, 0.9<br><br>**Surface reflectivity:**<br>Lambertian albedo at 0.04 |

VLIDORT models radiative transfer processes at a specific wavelength in a stratified atmosphere. It requires geometrical and "optical" information about the atmospheric layers and the underlying ground surface. These include layer heights, pressure and temperature at layer boundaries for refractive geometry calculations, solar zenith, viewing zenith direction and relative azimuth angles between the viewing direction and solar position. Each atmospheric layer is described by total optical thickness, total single scatter albedo, and the set of Greek matrices specifying the total scattering law.

VLIDORT simulations were performed for the US 1976 standard atmosphere divided into 67 layers (same as in Frieß et al., 2019) with 0.1 km layers from the surface to 4 km; 0.5 km layers from 4 to 8 km and varying





width up to 60 km. Since surface reflectivity has a small effect on ground-based MAX-DOAS measurements we performed simulations only for a single Lambertian albedo of 0.04. Absorption only by two gases was considered in this study: ozone and $O_2O_2$. Light polarization, direct beam refraction, and inelastic scattering were not included in this study. Table 1 summarizes VLIDORT inputs and general settings.

Aerosol types in this study are described by a single scattering albedo and asymmetry factor combination with total 20 "types": (1) Single scattering albedo: 0.775, 0.825, 0.875, 0.925, 0.975; (2) Henyey-Greenstein asymmetry factor: 0.675, 0.725, 0.775, 0.825. Aerosol extinction coefficient profiles were generated by combining an exponential function at the surface with a "sliding" Gaussian function above. The aerosol total optical depth was partitioned between the exponential and Gaussian functions. Total AOD cases included

0.15, 0.3, 0.45, 0.6, and 0.75 with exponential to Gaussian partitioning fractions of 0.3, 0.6 and 0.9. The Gaussian function peak center height was varied from 0.5 km to 2 km in steps of 0.5 km. The Gaussian function peak width was varied too: 0.1, 0.2, 0.3, and 0.5 km. This results in 4800 aerosol cases and a total of 1459200 measurement simulations (sky scan). While VLIDORT simulations were performed for an atmosphere divided into 67 layers ML training was done by resampling onto 23 layers only. The new layer

heights are: 100 m from the surface to 1km, 200 m from 1 km to 3 km, 500 m from 3 km to 4 km, and the last layer is 56 km high. The new layer partial AODs were generated by adding the neighboring layer partial aerosol optical depths. ML algorithm was trained on 75% randomly selected measurement simulations (1094400 samples) and evaluated on the remaining 25%.

## 5. Learning inverse mapping using ML

A novel multi-output sequence-to-sequence model to learn the inverse mapping from MAX-DOAS measurements to aerosol optical properties is illustrated in Figure 3. The inputs to the ML model comprise of a sequence of $\Delta AMF^{aerosol}$ at 16 VZAs, as well as two scalar inputs: the solar zenith angle (SZA) and the relative azimuth angle (RAA) of the single sky scan. To extract sequence-based features from MAX-DOAS data, a 1-dimensional Convolutional Neural Network (CNN, Fukushima, 1980; LeCun et al., 1999) applies

a learnable convolution operator on the sequence of MAX-DOAS values to produce a sequence of hidden features. These hidden features are then fed into the subsequent neural network layers to predict a sequence of partial AOD values, along with two scalars: single scattering albedo and asymmetry factor. Given the sequential nature of the partial AOD outputs, we employ a Long Short-Term Memory network (LSTM, Hochreiter and Schmidhuber, 1997) that is able to capture varying scales of memory in the sequence of partial

AOD values at varying levels. The complete ML model is implemented in the Jupyter Notebook by using the Keras library. RMSprop was chosen as the optimizer and the mean squared error as the loss function (Hinton, 2012).



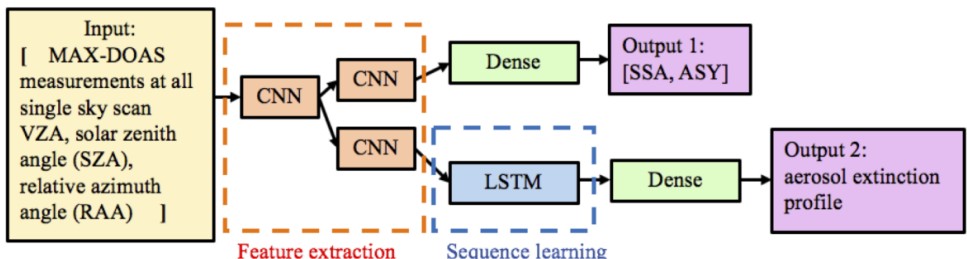

**Figure 3. Schematics of the multi-output sequence-to-sequence model for deriving aerosol optical properties from MAX-DOAS measurements.**

## 6. Results

Evaluation of the accuracy of ML mapping rules derived during the training stage for MAX-DOAS data inversion was done by comparing the "true" atmospheric aerosol properties to the ML inverted properties. The evaluation data set consists of 364800 MAX-DOAS simulated sky scans that are outside of the training set. The number of simulations in the evaluation data set as a function of solar zenith angle (SZA) and relative azimuth angle (RAA) are shown in Figure 4. Between 1100 and 1300 aerosol scenarios are present in each SZA-RAA bin.

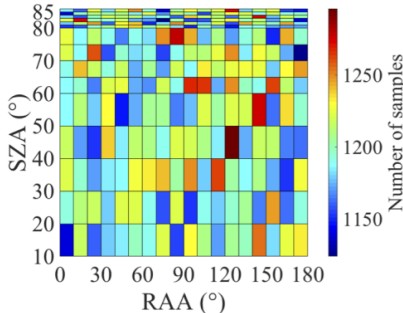

**Figure 4. Number of simulations in the evaluation data set as a function of solar zenith (SZA) angle and relative azimuth angle (RAA).**

The following ML predicted aerosol properties were evaluated: (1) asymmetry factor, (2) single scattering albedo, (3) total aerosol optical thickness, and (4) partial aerosol optical thickness for each layer from 0 to 4 km. A relative error $\epsilon$ of the retrieved by ML parameter $\hat{x}$ relative to the "true" value $x$ is calculated according to Eq. (10):

$$\epsilon \equiv \frac{\hat{x}-x}{x} \cdot 100\% \,, \tag{10}$$

The relative error evaluation presented in the subsequent sections was performed on the retrievals from a single ML training. Since ML itself introduces randomness during the training stage, we retrained the model 20 times with the same hyperparameters for evaluating the uncertainty of the ML training.



### 6.1. Asymmetry factor at 360 nm

The ML-based approach shows an ability to invert aerosol asymmetry factor with a mean error of -0.14% and two standard deviations of 2.04% and nearly normal error distribution (Fig. 5(a)). To evaluate if any dependence of the asymmetry factor retrieval exists on SZA and RAA the mean error and the two standard deviations are shown in Fig. 5(b, c). These distributions suggest that dependence of the asymmetry factor retrieval on SZAs and RAAs is relatively small. However, systematically higher relative errors are observed

around SZA of 65° and RAA of 30-40°. The cause of these elevated errors is not clear at this point.

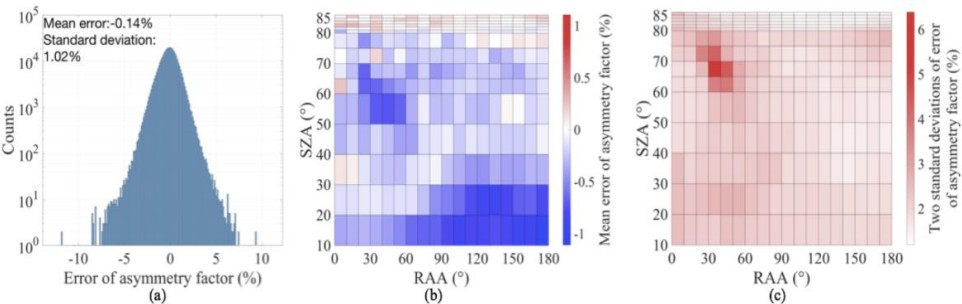

**Figure 5.** Asymmetry factor retrieval errors: (a) error histogram; (b) mean error as a function of SZA and RAA; (c) two standard deviations as a function of SZA and RAA.

### 6.2. Single scattering albedo at 360 nm

Similar high accuracy is achieved for ML retrieval of the single scattering albedo with a mean error of 0.19% and two standard deviations of 3.46% and nearly normal error distribution, somewhat positively skewed (Fig. 6). Slightly higher errors are observed at RAA smaller than 60° and most SZA.

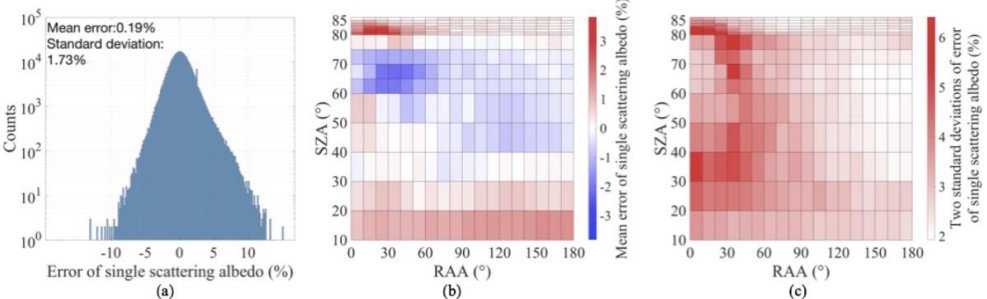

**Figure 6.** Single scattering albedo retrieval errors: (a) error histogram (b) mean error as a function of SZA and RAA (c) two standard deviations as a function of SZA and RAA.

Mean errors are also larger at small RAA and SZA > 85°. Traditional optimal estimation techniques also struggle with the MAX-DOAS data inversion at small RAA due to uncertainty in aerosol forward and backward scattering.

### 6.3. Total aerosol optical depth at 360 nm





Total AOD retrieval is more challenging for the ML model than the single scattering albedo or asymmetry factor, especially at lower total AOD levels. Box plots of the total AOD error for different "true" total AOD values are given in Fig. 7. In general, ML algorithm tends to underestimate total AOD from the mean error ± 2 standard deviations of -8.39 ± 8.81% (total AOD 0.15) to -1.52 ± 3.10% (total AOD of 0.75). Total AOD retrieval error distribution over all cases is close to Gaussian distribution, but with two peaks (Fig. 8).

The mean error (± two standard deviations) is -3.58% ± 7.68%. The bias of the model does not have much dependence on SZAs and RAAs (Fig. 8(b)). Still, lager errors and uncertainties can be observed at higher SZAs and lower RAAs (Fig. 8(c)).

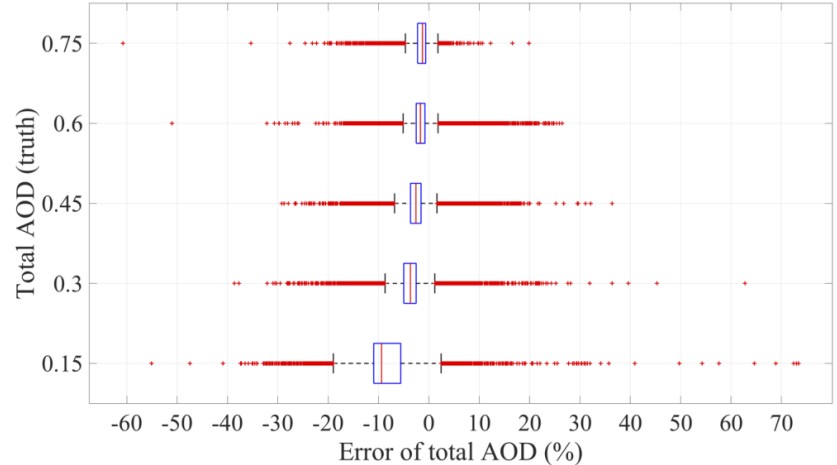

**Figure 7. Box plots of total AOD prediction errors for each "true" total AOD value.**

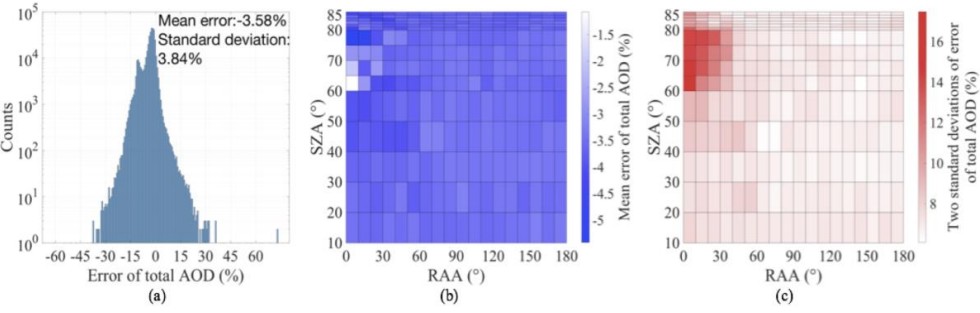

**Figure 8. Total AOD retrieval errors: (a) error histogram (b) mean error as a function of SZA and RAA (c) two standard deviation as a function of SZA and RAA.**

### 6.4. Partial aerosol optical depth profile from 0 to 4 km

The contribution of partial AOD retrieval error at each atmospheric layer from 0 to 4 km to the total AOD is shown in Fig. 9. This error contribution to the total AOD error depends on the absolute amount of aerosols and its altitude and on average is less than 1% per layer. Just like OEM methods, the ML method has lower


accuracy of retrieving elevated aerosol layers especially corresponding to smaller total AOD. The larger
distribution of relative errors in partial AOD at 1.5 km and 2 km is mainly due to the presence of elevated
layers in the training data that peaked at those heights. If the aerosol were also present in meaningful amounts
above those altitudes the error distribution would have been larger above 2 km.

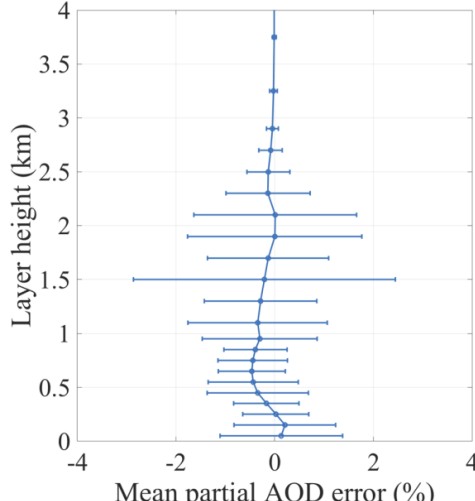

**Figure 9. Mean partial layer AOD error ± one standard deviation.**

A linear regression analysis of the "true" versus the retrieved partial AOD was performed using the least-
320 squares fitting for each layer from 0 to 2.2 km (Fig. 10). Intercepts of linear regression analysis for all layers
were zero with RMS $\leq 0.01$. High $R^2$ values (0.93 – 0.99) and slopes (m) close to one suggest that the ML
method relatively accurately estimates partial AOD at the layers between 0 and 2.2 km. As was noted earlier
lower retrieval accuracy is observed at the higher altitudes.





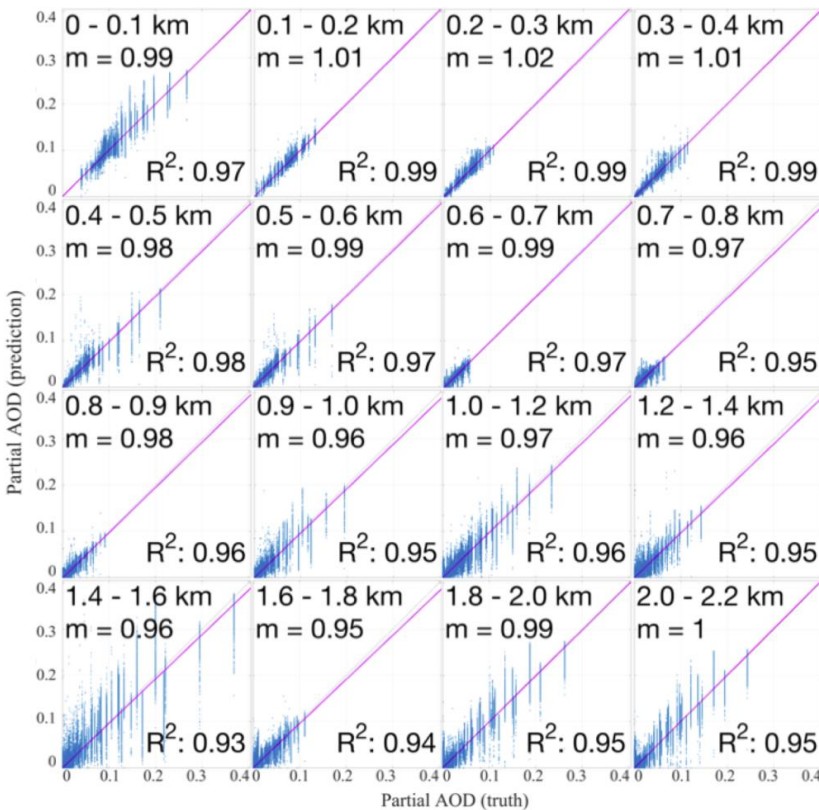

**Figure 10. Correlation between the retrieved partial AOD and the "true" partial AOD for each layer from 0-2.2 km ($retrieved\ partial\ AOD = m \cdot "true"\ partial\ AOD + intercept$). The intercept of all linear regression analyses is 0 with RMS < 0.01.**

Figure 11 shows some examples of the partial AOD profiles retrieved by the ML inversion model. Panels (a)-(h) in Fig. 11 contain randomly selected profiles out of the tested pool. While panels (i)-(l) contain some of the worst predictions. These examples show that the ML model is able to predict the elevated aerosol layers and even in those cases having large discrepancies, the model is still capturing the correct shape.

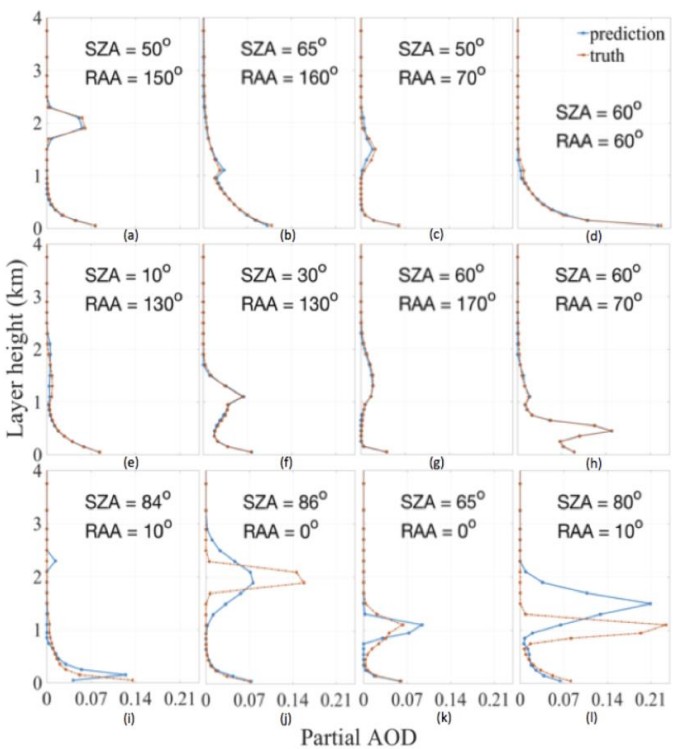

**Figure 11. Examples of predicted partial layer AOD profiles: (a)-(h) randomly selected examples and (i)-(l) bad predictions**

**6.5. Effect of random noise in ML training on the retrievals**

To estimate retrieval uncertainties due to random noise in ML training on the aerosol properties we reran the ML training stage 20 times. Mean errors and standard deviations for total AOD, single scattering albedo and asymmetry factor for each trained model are shown in Fig. 12.

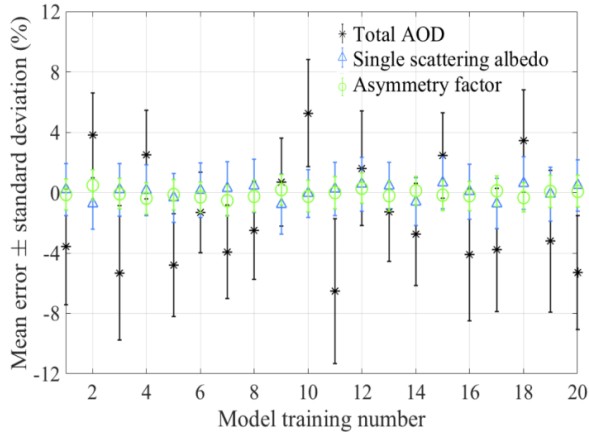

**Figure 12. Effect of random noise in model training on the retrieved aerosol properties**



Table 2 summarizes the effect of random model training noise on the retrieved properties. In general, most ML models result in a normal distribution of errors with an additional bias in the mean. Since the individual model training has a very small effect on error distribution (small changes in standard deviation between the different training runs) we add the variation in bias with standard deviation in quadrature to estimate the total error of the ML model including the random error of the training as:

(1) Total AOD error ± 2 standard deviations = -1.4 ± 10.1 %;

(2) Single scattering albedo error ± 2 standard deviations = 0.1 ± 3.6 %;

(3) Asymmetry factor error ± 2 standard deviations = -0.1 ± 2.1 %.

**Table 2. Statistics of aerosol property error analysis from 20 ML models (20 different training runs)**

| Optical property | bias ± std, % | Standard deviation ± std, % |
|---|---|---|
| Total AOD error | -1.43 ± 3.54 | 3.56 ± 0.64 |
| Single scattering albedo error | 0.06 ± 0.47 | 1.72 ± 0.10 |
| Asymmetry factor error | -0.08 ± 0.25 | 1.01 ± 0.03 |

**7. Conclusions and future work**

This paper presents a fast ML-based algorithm for the inversion of $\Delta SCD(O_2O_2)$ from a single MAX-DOAS sky scan into aerosol partial optical depth profile, single scattering albedo and asymmetry factor at 360 nm. Training and evaluation of ML algorithm are performed using VLIDORT simulations of $\Delta AMF(O_2O_2)$ for about 1.45 million scenarios with 75% randomly selected cases for training and 25% (~ 365 thousand cases)

for evaluation.

Evaluation of four retrieved aerosol properties (asymmetry factor, single scattering albedo, total AOD and partial AOD for each layer from 0 to 4 km) shows good performance of the ML algorithm with small biases and normal distribution of the errors. 95.4% of the retrieved optical properties have errors within the following ranges: (-1.4 ± 10.1) % for total AOD, (0.1 ± 3.6) % for single scattering albedo, and (-0.1 ± 2.1) %

for asymmetry factor. Linear regression analysis using the least-squares fitting method between the "true" and retrieved layer partial AODs resulted in high correlation coefficients ($R^2 = 0.93 - 0.99$), slopes near unity (0.95 – 1.02) and zero intercepts with RMS ≤ 0.01 for each layer from 0 to 2.2 km. The ML algorithm, in general, has less accuracy retrieving low total AOD scenarios and their corresponding profiles. Even in those scenarios with less accuracy, the ML model is still capable of capturing the correct profile shape.

Application of ML-based algorithm to real data inversion has the following advantages:

(1) Fast real-time data inversion of the aerosol optical properties;

(2) Simple implementation by using an HDF file with the model coefficients in open source codes such as Python;

(3) Ability to retrieve single scattering albedo and asymmetry factor;



(4) Use of the ML algorithm retrieved aerosol extinction coefficient profiles; single scattering albedo and asymmetry factor as initial guess inputs in more formal inversion algorithms (with radiative transfer simulations).

To verify that the ML retrievals are representative of the physical processes we suggest simulating $\Delta SCD(O_2O_2)$ using a radiative transfer model (e.g. VLIDORT) with the ML retrieved properties as inputs

(aerosol extinction coefficient profile, single scattering albedo, and asymmetry). Deviations from the measured and simulated $\Delta SCD(O_2O_2)$ should be included in error analysis.

To make the ML model more robust the training data should include more realistic aerosol inputs and radiative transfer simulations including 1) Rotational Raman scattering simulations to add Ring measurements from MAX-DOAS; 2) different surface albedos; 3) more realistic aerosol profiles (e.g. from

a 3-D multi-wavelength aerosol/cloud database based on CALIPSO and EARLINET aerosol profiles, LIVAS (Amiridis et al., 2015)); 4) multiple wavelengths.

**Code/Data availability**

All data used in this study (radiative transfer simulations and ML model from a single training) are available from (Dong et al., 2019).

**Author contribution**

Elena Spinei conceived the original idea of the algorithm and performed radiative transfer simulations to generate training and test data sets. Yun Dong developed the machine learning (ML) algorithm, conducted training and data inversion, performed error analysis and visualization. Anuj Karpatne guided the design of the ML model architecture. Elena Spinei supervised the project. All authors discussed the results and

contributed to the final manuscript.

**Competing interests**

The authors declare that they have no conflict of interest.

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
