# Peer review of "A feasibility study to use machine learning as an inversion algorithm for aerosol profile and property retrieval from multi-axis differential absorption spectroscopy measurements."

_Atmospheric Measurement Techniques, 2019_

## Referee Comment (RC1) · Anonymous Referee #1 · 26 Nov 2019

The manuscript "Machine learning as an inversion algorithm for aerosol profile and property retrieval from multi-axis differential absorption spectroscopy measurements: a feasibility study" presents a machine learning (ML) approach for aerosol retrieval from multi axis differential optical absorption spectroscopy (MAXDOAS). This is a new and innovative approach for aerosol retrieval from MAXDOAS measurements an therefore deserves certainly publication.

The paper gives a brief introduction to inverse modelling in general (sufficient or maybe even too much), to the MAXDOAS technique (sufficient), aerosol characterization (too

much, this is in the introduction and in the description of training data creation) and describes to some (sufficient) extend the forward model (VLIDORT 2.7) used to create the training dataset. However, it only describes very briefly the ML technique in general (not sufficient) and includes hardly any details about the actual neural network (architecture+hyper parameters) that was trained, nor does it include details about the training. The authors claim that the manuscript describes the new ML approach, but in fact, more space is used for the description of the training data set creation and about aerosol phase functions than on the ML itself. This does not reflect the title.

I certainly recommend this manuscript for publishing because it contains an innovative and very promising approach. I also think that the general approach to the work is good and that the results are of great interest and importance. Since there is no "moderate revision", I select "mayor revision", however, it is really just a bit more than "minor revision".

I strongly recommend two mayor changes:

(I) The weight of the manuscript needs to be on the ML approach, this is currently not the case. A general short description of neural networks is needed. Some details about the software package used, explanation of the choice of network architecture, why CNN why LSTM, the detailed settings etc (see SPECIFIC COMMENTS for suggestions and questions about the settings)...

(II) While the first point is likely just a matter of adding some more content to the manuscript, the second point weights a bit heavier and might include some more work: Normally, for ML, the data is split into three sets: (1) a training dataset (2) an evaluation dataset used during training to identify when the training results in overfitting (3) a completely new set of data for testing. I do not see that this has happened in this study, this is a huge weakness of the approach. The authors seem to have only used a validation data set (25% of the total data set) for the testing but no proper testing with parameters outside the training range (so not only "not this specific combination") was

performed.

GENERAL COMMENTS:

(1) Introduction

There is a lengthy (and maybe not super accurate, see below) description on the aerosol phase function and the asymmetry parameter, both in the introduction and in Sect.4. However, there is no information on ML in the intro. This does not at all reflect the title. After all, this manuscript claims to be about the ML as inversion algorithm. Suggestion: Bundle the aerosol information from here and from Sect. 4 in the section about training/validation/test data creation (which is currently Sect. 4) and include some paragraph or two on ML use in inverse modelling and general ML.

(2) Ordering of content

I suggest a different ordering: (1) general introduction including advances in ML, aerosol importance in general, current retrieval techniques and why ML should be applied to aerosol retrieval (2) MAXODAS method description (3) Aerosol properties and modelling and forward modelling with VLIDORT (4) Overview of the methodology of the 3 necessary steps (instead of selling it as two steps as done in Sect. 3, where the first of the two has itself two steps) and a detailed description of the specific ML setup and choice of hyperparameters. (5...) as before

(3) Sect.2:

While what is written about OEM and parametrized methods is true, most of it is true for ML as well (i.e regarding e.g. the T/P profile). This section paints an overly dark image of OEM and parametrized codes. I think that the main problem with "traditional" methods is indeed the time they take, and this should be clearly (even more clearly) stated, since this is the one huge huge advantage of using ML. Also, especially around line 136, it gives the impression of full profile retrieval of asy and ssa, while in fact, it is "only" the aod profile and single scalar values asy and ssa valid for all layers.

(4) content of Sect. 3 + 5:

Judging from the title, this should be the heart of the manuscript. It is not. Also, why this strange split of the topics? Does it make sense? As mentioned under (2), I would keep a very rough overview of the three steps in the introduciton and put all the remaining info on ML in Sect. 4. Some comments what is missing:

(a) which backend was used? Tensorflow, Theano? Some other? Why the mentioning of the yupiler notebook? Why was it used at all? Certainly no web-based interactivity is needed? Why wasn't it simply put in a plain python script?

(b) CNN is normally used in ML for image recognition, why is it used here? Is the CNN here to correctly identify the fixed parameters RAA and SZA (which have in fact a 1:1 mapping between input and output)? Why was it decided to split for profile and ssa/asy retrieval?

(c) Why is LSTM used? Maybe some intro on recurrent neural networks in general is needed. This seems to indicate, also from the bit of explanation that comes with it, that scans are not considered separately, but as a function of time.... (so a scan from now and then from 10 minutes, not one from here and now and the next one from tomorrow and somewhere else). However, this seems not to fit your introduction and abstract where you very specifically write about a single scan. This is very confusing and needs explanation. Also maybe, you can start with explaining what a SimpleRNN layer is and why this was not chosen?

(d) What were the choices of the hyperparameters? Which batch size was used? Which lr was used for the RMSprop? Where there any drop out layers? Which activation function was used? There is no information on any of the parameters. How many nodes do the layers have?

(5) what happens if the network gets data that is by no means covered by the training data (i.e. completely outside the range in one or more parameters?) What is the

effect of measurement noise (also including "noise" from situations that are not 1 dimensional)?

SPECIFIC COMMENTS:

(1) page 1, line 23 "... and have relative short lifetimes..." –> relative to what? Also, few minutes to few weeks spans about 5 order of magnitude in time, while one end of this span can be considered as short, the other cannot really. Please specify "relative".

(2) page 1, line 26: apart from all the properties already listed, what else do you mean with "physical properties" as opposed to optical? This is very unclear

(3) page 1, line 28 "The spatial and temporal distribution of aerosols ... is greatly affected by ... the type of aerosols". I think this is incorrect, the correct verb here is "depends on".

(4) page 2, line 39–40: If you put this statement, then you need to explain more. I also cannot see any connection of this statement to the rest of the paper. The minimum that should be added is how it depends on the surface albedo.

(5) page 2, line 41–42: "escpecially of anthropogenic origin" "of"? or "for"? This sentence does not make too much sense like it is, reformulation needed.

(6) page 2, eq2 and eq3: I would think that the range of the asymmetry parameter as such depends on the normalization of the phase function, so you need to have integral(phasefunction) over 3D angle = 1. If so, then the first moment <cos theta> is the asymmetry btw. forward and backward scattering. So with this, would you not have a factor of 1/4pi missing in the HG phase function? Maybe you could check the normalization factors for consistence btw. g and P.

(7) page 2, eq. 4: You seem to use tensor notation to make a difference btw. covariant and contravariant tensors and apply Einstein summation convention. However, you still put the summation sign, but without indicating what you are actually summing over.

(8) page 4, line 101: "approximately known"? Please clarify.

(9) page 5, around line 136: Since it was highlighted before that

(10) page 5, line 153..154: both input and output states run to N, one of them should have a different limit, maybe... M? Otherwise it is confusing, especially because it is written that x has 67 layers, but y has "only" 16 angles.

(11) page 7, line 196: Although VLIDORT has as direct input the viewing zenith angle, most people in the MAXDOAS community are more familiar with the elevation angle. Maybe it is an idea to change this to make it easier to connect to.

(12) page 8, line 199, 201, and other listings in the text of parameters that are summarized in the Table 1: I do not think that they need to be repeated, I think it is enough if they are in the table.

(13) page 8, table 1: Can you comment on how the direct sun cases for raa=0, sza=vza are handled?

(14) page 9, line 223: why do you need ozone absorption?

(15) page 9, line 230: maybe a small sketch to explain the aerosol profile parameters (with the two components of the profile) would be helpful

(16) page 9, line 37: The 25% were fixed between the 20 realizations, or not? Also, did you monitor the evaluation loss? It would be really good to see some plots here of the evaluation loss as a function of epoch. Also, please comment on how over-fitting was mitigated.

(17) page 9, line 236: this height is the middle height or the height of the upper boundary? This is not clear.

(18) page 9, line 247ff: I would certainly describe the architecture of the network here, not only the Fig. 3. Also, dense layers are not explained. Also, how many nodes in the layers? Do you use maxpooling layers btw. your conv1d layers? What is the size

of your convolution window? And again, how was the architecure chosen? Why does it make sense to separate the SSA and the ASY the way you do? Do you extract the SZA and RAA as well? They should certainly be == the input? Is there a test on this?

(19) page 9, line 259: While you do use 25% for test (or do you actually use this for evaluation? Not really, because you use it to test the network. What was used for the evaluation then?), because you use the same type of parametrization, this is not a good test. A different, unseen set of data should be used. How do features that were not included in the training dataset at all (by all means outside the parameter range) affect the result? What about thin cloud layers above 4 km, do they affect the result? The tests included here are not very useful.

(20) page 10, line 275 & 285: given the range of parameters, using eq. 270, the maximum error is about 20%, not 100%. This puts these low numbers in context.

(21) page 11, Fig. 5: When you wrote earlier that the mean error is "-0.14", you really just took the mean over all angles? What is the significance of this? If it were in have the parameter space +50% and in the other -50%, its mean is still 0... and the model really not good, so what is the significance of the mean error here?

(22) page 12, Fig. 7: please explain the box whisker plot. Is the line mean or median? The box is how much percentage? The whisker? There are different conventions...

(23) page 16, line 351: This paper does not present the ML based algorithm. It presents some of the results and that's it. There is not enough information on the ML model. This sentence is not summarizing the paper.

(24) page 16, line 363: maybe this is because of the choice of training paramters as a linear distributed AOD?

TECHNICAL CORRECTIONS AND SUGGESTIONS

Since I expect some mayor change in the text, only a few comments here:

(1) Many times, there are definite articles missing (e.g. page 3 line 65 "The MAX-DOAS..", page 3 line 84: "The DOAS technique", page 4 line 91 "The offset term...")

(2) Eq. 5 on page 4 is not referred to in the text.

(3) page 7, line 204: I highly doubt that Clemer et al 2010 is the only code here. I would add a "e.g.".

(4) page 9, line 229: I think you miss the AOD=0 case in this list

(5) page 10ff: I suggest to use an equi-distant grid for the raa-sza plots, as they are now, it gives a biased impression to the eye.

(6) page 12, line 311 f: I cannot quite understand the sentence "This error contribution..." maybe you can reformulate

---

## Referee Comment (RC2) · Anonymous Referee #2 · 3 Dec 2019

Dong et al. introduce a new MAX-DOAS aerosol inversion algorithm which is based on machine learning. The algorithm itself utilizes neural networks which were trained with a pre-calculated synthetic dataset of $O_4$ AMF with various profiles, geometries and aerosol properties. A fraction of this dataset is used for the validation of the algorithm by analyzing results based on the difference between true and retrieved quantity.

This novel approach seems to be promising, particularly due to the advantage of being able to retrieve single scattering albedo (SSA) and asymmetry factor along with profiling information in near real-time.

[Figure]

**General comments**

The manuscript is well written but it lacks necessary information and analyses. I suggest to change the manuscript according to the following points:

1. Not enough information about the machine learning (ML) algorithm itself are given. The introduction focuses on aerosols and MAX-DOAS only without introducing ML. In section 5, there is no explanation why the individual ML steps were chosen as they are.

2. The validation section appears to be insufficient to assess the performance of the algorithm:

   (a) Why not changing the testing dataset to realistic profiles which are not included in the training data? How can you be sure that you do not over-fit your results?
   (b) Why not using larger aerosol loads?
   (c) Why did you use 16 different elevation angles for the testing dataset even though this number is much too high for most measurement locations? What happens if you just use 8 or 10 elevation angles? Does the algorithm still performs well?
   (d) The training dataset was created by using an US standard atmosphere. This is mostly a poor representation of the true atmosphere. What happens if the conditions change?
   (e) Why not testing the algorithm on real data?

3. Are there plans to extend the training dataset by more wavelengths, SSA/asymmetry factors, albedo, profiles, trace gases (as suggested in Sec. 7)?

[Figure]

4. Note that many articles are missing in the manuscript.

**Specific comments**

**P2, L36-42:** There is no need to show the equation of SSA and its detailed description. I suggest to change those 6 lines into one sentence only.

**P2, L43-62:** This part about the aerosol phase functions is much too long, especially since there is no further discussion about this topic in your paper. The lines about the Legendre expansion could be completely removed without loosing important information for the understanding of the manuscript.

**P2, L58:** When kept, please change the index L of $P_L(\cos\theta)$

**P3, L65:** "The" MAX-DOAS...

**Figure 1:** A single sky scan...

**P3, L84:** "The" DOAS technique...

**P4, L91:** "The" offset term...

**P4, L101:** "Forward model parameters that are considered approximately". I guess you are referring, among other parameters, to a priori knowledge when using "approximately" here? Please change the wording or reformulate this sentence.

**P4, L114-115:** "A priori information about...". I find this sentence to be rather confusing. What do you want to say here?

**P5, L123-125:** "None of the algorithms perform perfectly". That depends on what you understand as "perfect". I don't think that it is possible at all to retrieve the true atmosphere in a extremely high vertical resolution. However, as far as I know, the second part of this sentence is correct. I would suggest to reformulate the first part.

**P5, L123-130:** These 7 lines describe problems that also apply to your algorithm and only highlight on problems of existing algorithms. As long as you don't show perfectly validated profiles from real measurements, the reader does not believe that your algorithm "performs perfectly".

Note that you also used external information about the atmosphere for creating your training dataset. You directly applied a priori knowledge by using exponentially decreasing profiles including Gaussian's for your dataset. And I would not say that a priori knowledge does not exist. You look out of the window and know that it is a hazy day so you adapt the a priori. Sometimes you know about local sources or have ancillary measurements available. I would strongly suggest to change this paragraph.

**P7, L176:** AMF represents → An AMF represents

**P7, L178:** observations → observation

**P7, L181:** Where "the" vertical column density...

**P7, L183-191:** This part could be shortened as you have already introduced aerosols before.

**P7, L205:** "The" VLIDORT code...

**Table 1:** Why do your Gaussian profiles don't have center heights higher than 2km? Since the vertical sensitivity for higher altitudes is an issue for common algorithms, I am wondering if your algorithm performs better here? What is the scaling height of your exponential functions?

**P9, L235:** What was the reason for changing the grid step width to a coarser resolution for higher altitudes? I have the feeling that your choice of Gaussian profile center heights and retrieval grid steps might deteriorate retrieval results for higher altitudes (as indicated in Fig. 9 and 11).

**Section 5:** I think it would be nice to add more information to this section to explain also the in-between steps and parameters of your CNN and LSTM.

**P9, L251:** "RMSprop was chosen...". Please explain.

**Figure 4:** It would be interesting for the reader to see a similar plot describing the profile shape distribution. You could show 3 more plots for different partitioning, showing Gaussian center heights and width on x and y axis, respectively. Furthermore, the number of profiles with a certain total AOD would also be interesting (especially when looking at Figure 7). I fear that the reader might loose the connection to the actual profile shape due to the rather statistical analysis in the following paragraphs.

**Figure 5, 6 and 8:** It is interesting to see that mean error and standard deviation show areas with high or low values at certain geometries. I was wondering if this is a matter of the scattering angle (angle between incident and outgoing photon assuming single scattering)? Could you please create a plot showing the scattering angle versus the respective error/standard deviation? Since e.g. RAA = 30° and a low SZA is equivalent to a large scattering angle (e.g. Fig 5c) it might show issues for certain scattering geometries. In addition, you could check if there are certain profile shapes or aerosol parameters more frequent for areas with a large standard deviation or high mean errors compared to other geometries. I was also wondering about the outliers in all three histograms. Any reason for that?

**P11, L292:** I agree that OEM methods also struggle with data inversion measured at small RAA but I was wondering why your synthetic analysis fails?

**P12, L295:** "The total AOD retrieval..." or "The retrieval of total..."

**P12, L297:** In general, "the" ML algorithm

**P12, L299:** What is the reason for the second peak in the histogram?

**Figure 7:** Please explain all depicted quantities (mean, median, percentiles...) in the caption of this figure. Here, it would also be interesting to see if the largest underestimations correspond to certain profiles or parameters.

**Figure 9:** Why is the error larger for 1.5km than for 2km? Since you also included Gaussian's with Peak heights around 2km, I would expected the largest error at higher altitudes. Especially when considering the higher sensitivity of MAX-DOAS measurements for aerosol loads closer to the surface (which can be seen for altitudes lower than 1.5km).

**Figure 10:** It appears that there is also an underestimation of the predicted AOD for all sub-figures with true partial AOD's larger than 0.2. For example in the upper left sub-figure, but also in the second row (first figure), the third row (2nd and 3rd fig). Do these underestimations correspond to problematic scenarios/geometries/parameters?

**P16, L353:** Training and evaluation of "the" ML

**P16-17, L365-372:** Points 1 and 2 are valid but only a demand for near real-time applications. I doubt that there is a need for science to have profiles immediately after the measurement. For point 3, I was wondering if this is a major advantage. The dependence of profiling results on SSA is rather small and the Henyey-Greenstein approximation is in most cases a poor representation of the scattering distribution of aerosols. So why should the reader decide for your algorithm when an AERONET station nearby measures "real" phase functions and SSA? In point 4 you even diminish the potential of your approach by saying that it might be used as an initial guess for other algorithms. To me, this does not sound as if the authors are convinced of the capability of ML algorithms. If this is true, why not? If not, why are there no strong arguments in favor of your approach?
* * *

---

## Author Response (AR1)

Dr. Omar Torres
Associated Editor
Atmospheric Measurement Techniques

5 Re: "Machine learning as an inversion algorithm for aerosol profile and property retrieval
from multi-axis differential absorption spectroscopy measurements: a feasibility study."
Amt-2019-368

Dear Dr. Torres,

Please see attached a revised version of Amt-2019-368, with the changes tracked by the
10 Microsoft Word. We have responded to the comments from the two reviewers.
Reviewers' comments are in bold italics; our responses are in regular Times font.

Thank you for your consideration.

Best Regards,
Yun Dong, Elena Spinei and Anuj Karpatne

**Responses to the comments by Reviewer 1.**

**Changes recommended:**

*(I) The weight of the manuscript needs to be on the ML approach, this is currently not the case.*

20   We have removed some details about the aerosols and have added introduction to
   machine learning to section 1. (lines 37 to 137 in the new version)

*(II) Normally, for ML, the data is split into three sets: (1) a training dataset (2) an evaluation dataset used during training to identify when the training results in overfitting (3) a completely new set of data for testing. ... The authors seem to have only*
25 *used a validation data set (25% of the total data set) for the testing but no proper testing with parameters outside the training range (so not only "not this specific combination") was performed.*

   We thank the reviewer for bringing this point on the correctness of our evaluation
   setup and we would like to clarify that no part of the test data was used in any
30   way during training, thus ensuring the validity of our test results in representing
   the performance of our ML model on samples outside the training set. Note that in
   any supervised ML experiment, it is very important to ensure that there is no
   overlap between the training and test sets, so that the performance on the test set
   is a true indicator of generalization performance, i.e., the performance on
35   "unseen" instances never seen before during training (also known as out-of-
   sample instances). This is generally done by holding off a fraction of the overall
   data during training, thus partitioning the overall data into two sets: a "training
   set" used only for model building, and a "test set" used only for model evaluation
   (for further details on evaluating supervised ML models, see Tan et al., 2018 and
40   Fiedman et al., 2001). A common approach for partitioning the overall data into

training and test sets is to consider random sampling, also known as the random holdout method [ak1]. In our experiments, we randomly partitioned the overall data into a training set (comprising of 75% instances) and a test set (comprising of 25% instances). Further, an optional procedure that is sometimes followed is to hold a certain fraction of the training set as the "validation set" and monitor the performance on the validation set during training to either avoid overfitting or to tune the hyper-parameters of the ML model. Since the validation set is used during model building (although indirectly) it is no longer considered as a representation of "unseen" instances, and hence, the validation performance is not a true indicator of generalization performance. Note that in our work, we did not make use of any validation set, as the values of all hyper-parameters in our ML model were kept constant across all experiments. Instead, we only report our results on the test set that was not used during training, either directly or indirectly.

We have added the following text to the revised paper to address this comment: *Section 4 (lines 315 to 318 in the new version):* "ML algorithm was trained on 75% randomly selected measurement simulations (1094400 samples) and model performance was tested on the remaining 25%. Note, that no validation data was held off from the 75% training set for tuning hyper-parameters of our ML model, as all ML hyper-parameters were kept constant across all experimental settings in this paper."

   *Section 5 (lines 378 to 385 in the new version):* "We trained the model on 75% of the dataset for 124 epochs with a batch size of 640. The following choice of hyperparameters was used: choice of optimizer=RMSprop, lr=0.001, rho=0.9, epsilon=None, and decay=0.0. We did not perform any hyper-parameter tuning on a separately held validation set inside the training set, and the values of all hyper-parameters in our ML model were kept constant throughout all experiments in the paper on the test set. In order to ensure that there was no overlap between the training and testing steps, we did not make use of the test data either directly or indirectly during the training phase, either for learning parameter weights or selecting hyper-parameters."

*General comments:*
*(1) There is a lengthy (and maybe not super accurate, see below) description on the aerosol phase function and the asymmetry parameter, both in the introduction and in Sect.4. However, there is no information on ML in the intro. This does not at all reflect the title. After all, this manuscript claims to be about the ML as inversion algorithm. Suggestion: Bundle the aerosol information from here and from Sect. 4 in the section about training/validation/test data creation (which is currently Sect. 4) and include some paragraph or two on ML use in inverse modelling and general ML.*

We've shorten the aerosol part and added general description of ML and detailed description of the ML model.  See reply to general comments (1)

 *(2) I suggest a different ordering: (1) general introduction including advances in ML, aerosol importance in general, current retrieval techniques and why ML should be applied to aerosol retrieval (2) MAXODAS method description (3) Aerosol properties and modelling and forward modelling with VLIDORT (4) Overview of the methodology of the 3 necessary steps (instead of selling it as two steps as done in Sect. 3, where the*

*first of the two has itself two steps) and a detailed description of the specific ML setup and choice of hyperparameters. (5...) as before*

See reply to general comments (1)

90 *(3) While what is written about OEM and parametrized methods is true, most of it is true for ML as well (i.e regarding e.g. the T/P profile). This section paints an overly dark image of OEM and parametrized codes. I think that the main problem with "traditional" methods is indeed the time they take, and this should be clearly (even more clearly) stated, since this is the one huge advantage of using ML. Also, especially*

95 *around line 136, it gives the impression of full profile retrieval of asy and ssa, while in fact, it is "only" the aod profile and single scalar values asy and ssa valid for all layers.*

We have reworded this part (*lines 203-206*):

"Aerosol extinction coefficient profiles are inverted while aerosol single scattering albedo and asymmetry factor are typically assumed based on the co-

100 located AERONET measurements. They also require external information about the atmosphere (e.g. temperature and pressure profiles) that might not be readily available at the measurement time scales, and a priori information that does not typically exist."

*(4)*

105 *(a) Which backend was used? Tensorflow, Theano? Some other? Why the mentioning of the yupiler notebook? Why was it used at all? Certainly no web-based interactivity is needed? Why wasn't it simply put in a plain python script?*

TensorFlow backend was used. We mention Jupyter Notebook just because the code is implemented in Jupyter Notebook. Yes, web-based interactivity is not

110 needed and of course we can use plain python script. We used Jupyter notebooks just for easy sharing of code, analysis of results, and reproducibility of experiments.

*(b) CNN is normally used in ML for image recognition, why is it used here? Why is LSTM used? Maybe some intro on recurrent neural networks in general is needed.*

115 *This seems to indicate … that scans are not considered separately, but as a function of time.... (so a scan from now and then from 10 minutes, not one from here and now and the next one from tomorrow and somewhere else). However, this seems not to fit your introduction and abstract where you very specifically write about a single scan. This is very confusing and needs explanation. Also maybe, you can start with explaining what*

120 *a SimpleRNN layer is and why this was not chosen?*

Different from image recognition in which 2D CNN is usually used, what we use in our model is 1D CNN which is good for capturing features from 1D-sequences. We've also added general introduction of LSTM. As mentioned above, we consider the profile as a sequence, that's the reason we use the LSTM. We do not

125 use the LSTM in a typical way where the input or output sequence is a time series. In our case, it is nothing related to time but a series of partial AOD values at sequential heights. Simple RNN is inadequate to capture long-term memory effects where the inputs-outputs at a given element of the sequence can affect the outputs at another element of the sequence separated by a long interval. Actually

130 we've tried simple RNN and it does not work as well as LSTMs.

We have added the following text to the paper in Section 5 (lines 330 to 340) to address this comment:

135 "Note, that in our supervised ML formulation, there are sequences in both the input signals and output signals, namely $\Delta AMF^{aerosol}$ sequence and partial AOD sequence, respectively. Further note that the input and output signals used in our problem setting are of very different types and thus have different dimensionalities (e.g., $\Delta AMF^{aerosol}$ takes 16 values at varying VZAs while partial AOD takes 23 values at varying atmospheric layer depths). We thus first apply a 1-dimensional CNN to extract features from the sequence part of the input

140 signals. Note that our input signals are not image-based, which is one of the common types of input data for which CNNs are used. Instead, our input data is structured as a 1D sequence, and the convolution operations of CNN help in extracting sequence-based features from the input signals that are then fed into subsequent ANN components. We also use an LSTM to model the sequence part

145 of the output signals. Note that our data contains no time dimension as we are only working with single scan data. However, it is the sequence-based nature of the output signals that motivated us to use LSTM models for sequence-based output prediction. Furthermore, the dataset we use for training is produced by a physical model (VLIDORT), where the relationship between the inputs and

150 outputs are known."

**(b) Why was it decided to split for profile and ssa/asy retrieval?**

We split profile and SSA/ASY retrieval because we consider the profile as a sequence (the partial AODs at adjacent layers are related) that needs to be

155 modeled using an LSTM, but the SSA/ASY are scalars that can be modeled using Dense layers. We've tried a lot of architectures and find that combining profile and SSA/ASY as a single output sequence results in inferior performance.
We have added the following text in the paper in Section 5 (lines 346 to 357) to address this comment:

160 "To extract sequence-based features from MAX-DOAS inputs, a 1-dimensional Convolutional Neural Network (CNN, Fukushima, 1980; LeCun et al., 1999) is first applied on the sequence of inputs (we concatenate $\Delta AMF^{aerosol}$ sequence with SZA and RAA to obtain an 18-length input sequence), which results in a sequence of preliminary hidden features. These preliminary hidden features are then sent to two

165 different branches of 1D-CNN layers that perform further compositions of convolution operators to produce non-linear hidden features for predicting two different types of outputs: (a) scalar outputs: SSA and ASY, and (b) sequence-based outputs: aerosol extinction profile. For the branch corresponding to scalar outputs, the features extracted from 1D-CNN layers are simply passed on to a fully-

170 connected dense layer to produce a two-dimensional output of SSA and ASY. For the branch corresponding to sequence-based outputs, the features extracted from 1D-CNN layers are fed to a Long Short-Term Memory network (LSTM, Hochreiter and Schmidhuber, 1997) to produce a sequence of partial AOD values at varying atmospheric layers."

175 **(c) What were the choices of the hyperparameters? Which batch size was used? Which lr was used for the RMSprop? Where there any drop out layers? Which activation function was used? There is no information on any of the parameters. How many nodes do the layers have?**

We've added a plot of the detailed architecture of the ML model in the
supplement with all the information.

180 We have also added the following text in the paper in Section 5 (lines 375 to 380)
to provide more details about the hyper-parameters of our model:

"RMSprop was chosen as the optimizer and the mean squared error was used as the
loss function (Hinton, 2012). We trained the model on 75% of the dataset for 124

185 epochs with a batch size of 640. The following choice of hyperparameters was used:
choice of optimizer=RMSprop, lr=0.001, rho=0.9, epsilon=None, and decay=0.0."

***(5) what happens if the network gets data that is by no means covered by the training
data (i.e. completely outside the range in one or more parameters?) What is the effect
of measurement noise (also including "noise" from situations that are not 1***

190 ***dimensional)?***

Though the outputs of the test set are not outside the range of the training data,
however, the mappings contained in the test set are different. And different
combination of SSA/ASY/profile produces different values of radiance. The
model hasn't seen these input values and output combinations of

195 SSA/ASY/profile before. As for the point you mentioned here, there are next
steps of our work. ML itself is a technique learning from the statistics of the data.
If applying on the dataset which is too different from the training set, of course
with high probability it cannot provide reliable predictions. The more the ML
model 'see', the better it works. Thus, we need to include more realistic aerosol

200 inputs and radiative transfer simulations as mentioned in the 'Conclusion and
Future Work' section. We will also consider noise in future work. For this work,
our key point is the 'feasibility', which aiming at demonstrating that it is feasible
to use ML technique into MAX-DOAS aerosol retrieval.

***Specific comments:***

205 ***(1) page 1, line 23 "... and have relative short lifetimes..." –> relative to what? Also,
few minutes to few weeks spans about 5 order of magnitude in time, while one end of
this span can be considered as short, the other cannot really. Please specify "relative".***

- we replaced "relatively short" with "variable".

***(2) page 1, line 26: apart from all the properties already listed, what else do you mean***

210 ***with "physical properties" as opposed to optical? This is very unclear.***

- we removed this sentence since it did not add any new information: "The aerosol
classification depends on the aerosol source, composition, size and number distribution,
aging processes, and optical and physical properties."

***(3) page 1, line 28 "The spatial and temporal distribution of aerosols ... is greatly***

215 ***affected by ... the type of aerosols". I think this is incorrect, the correct verb here is
"depends on".***

- we replaced "is greatly affected by" for "greatly depends on"

***(4) page 2, line 39–40: If you put this statement, then you need to explain more. I also
cannot see any connection of this statement to the rest of the paper. The minimum that***

220 ***should be added is how it depends on the surface albedo.***

- to shorten the aerosol discussion in introduction we removed line 37 – 63 on p.2.

***(5) page 2, line 41–42: "escpecially of anthropogenic origin" "of"? or "for"? This
sentence does not make too much sense like it is, reformulation needed.***

- to shorten the aerosol discussion in introduction we removed line 37 – 63 on p.2.

225 *(6) page 2, eq2 and eq3: I would think that the range of the asymmetry parameter as such depends on the normalization of the phase function, so you need to have integral(phase function) over 3D angle = 1. If so, then the first moment <cos theta> is the asymmetry btw. forward and backward scattering. So with this, would you not have a factor of 1/4pi missing in the HG phase function? Maybe you could check the*
230 *normalization factors for consistence btw. g and P.*
      - to shorten the aerosol discussion in introduction we removed line 37 – 63 on p.2.
*(7) page 2, eq. 4: You seem to use tensor notation to make a difference btw. covariant and contravariant tensors and apply Einstein summation convention. However, you still put the summation sign, but without indicating what you are actually summing*
235 *over.*
      - to shorten the aerosol discussion in introduction we removed line 37 – 63 on p.2.
*(8) page 4, line 101: "approximately known"? Please clarify.*
      - we added (e.g., temperature and pressure profiles from atmospheric sounding or models)
240 *(9) page 5, around line 136: Since it was highlighted before that*
      - not sure how to interpret this comment
*(10) page 5, line 153..154: both input and output states run to N, one of them should have a different limit, maybe... M? Otherwise it is confusing, especially because it is written that x has 67 layers, but y has "only" 16 angles.*
245       - we replaced y number of elements with M
*(11) page 7, line 196: Although VLIDORT has as direct input the viewing zenith angle, most people in the MAXDOAS community are more familiar with the elevation angle. Maybe it is an idea to change this to make it easier to connect to.*
      - we agree that "elevation angle" is a more familiar term but the MAX-DOAS
250 community is well aware of the zenith angle definition.
*(12) page 8, line 199, 201, and other listings in the text of parameters that are summarized in the Table 1: I do not think that they need to be repeated, I think it is enough if they are in the table.*
      - we replaced the exact listing with the following: "… and nineteen viewing zenith
255 angles between 0 and 89$^o$ (see Table 1). To ensure that the training dataset contains all observation geometries feasible for MAX-DOAS sky scans we have included: nineteen relative azimuth angles (0 to 180$^o$, 10$^o$ step), and twelve solar zenith angles (0 to 85$^o$, 89$^o$ see Table 1)."

260 *(13) page 8, table1: Can you comment on how the direct sun cases for raa=0, sza=vza are handled?*
      - it is not handled in any special way. We do recognize that no meaningful profile information is available from such geometries and the forward scattering has large uncertainties.
265 *(14) page 9, line 223: why do you need ozone absorption?*
      - strictly speaking we do not need ozone absorption, but since there is no harm in its presence we left it in.
*(15) page 9, line 230: maybe a small sketch to explain the aerosol profile parameters (with the two components of the profile) would be helpful*

270     - we added: "Figure 11 demonstrates the aerosol profile samples, where the near
surface aerosol partial optical depth profiles are described by the exponential function
and the layers aloft are described by the Gaussian function with various widths and
heights added to the exponential function profile."
*(16) page 9, line 237: The 25% were fixed between the 20 realizations, or not? It would*
275     *be really good to see some plots here of the evaluation loss as a function of epoch. Also,*
*please comment on how over-fitting was mitigated.*
        - we did not perform the hyper-parameter optimization in a formal way, so no
cross evaluation was done. However, we did monitor training loss and it converged. To
eliminate the confusion, we have replaced "evaluated" with "performance tested".
280     *(17) page 9, line 236: this height is the middle height or the height of the upper*
*boundary? This is not clear.*
        - we replaced layer "heights" with "depths"
*(18) page 9, line 247ff: I would certainly describe the architecture of the network here,*
*not only the Fig. 3. Also, dense layers are not explained. Also, how many nodes in the*
285     *layers? Do you use maxpooling layers btw. your conv1d layers? What is the size of*
*your convolution window? And again, how was the architecure chosen? Why does it*
*make sense to separate the SSA and the ASY the way you do? Do you extract the SZA*
*and RAA as well? They should certainly be == the input? Is there a test on this?*
        - We've added a plot of model architecture in the supplement.
290     *(19) page 9, line 259: While you do use 25% for test (or do you actually use this for*
*evaluation? Not really, because you use it to test the network. What was used for the*
*evaluation then?)*
        - Hyper-parameter selection was done offline. We have not seen a significant
difference between the hyper-parameter choices for the selected architecture and did not
295     include the cross-evaluation at all during the training so 25% of the data that the model
NEVER saw are used for testing the performance of the model.
*Because you use the same type of parametrization, this is not a good test. A different,*
*unseen set of data should be used.*
        - The model never saw the test data so the test is valid
300     *How do features that were not included in the training dataset at all (by all means*
*outside the parameter range) affect the result?*
        - Ideally, the final MAX-DOAS inversion algorithm based on ML would "see"
most of the possible ranges, for example, profiles from the LIVAS database, but optical
properties varied across all aerosol types for the same profile so the solution is more
305     reliable.
*What about thin cloud layers above 4 km, do they affect the result?*
        - Friess et al 2018 used NASA real-time aerosol retrieval algorithm that is the
        basis for the dAMF (-dAMF= AMF – $AMF_{Rayleigh}$) analysis in this study. It was
        shown that the method is not sensitive to the aerosol/cloud layers above 4 km. We
310     assume the same applies to the ML-based algorithm.
*The tests included here are not very useful.*
        - We disagree with this statement. Most studies evaluating the performance of
MAX-DOAS algorithms (e.g. Fiess et al, 2018) have significantly simpler and smaller
data sets from both profile variability, observation geometry and optical properties that
315     were tested in this study.

*(20) page 10, line 275 & 285: given the range of parameters, using eq. 270, the maximum error is about 20%, not 100%. This puts these low numbers in context.*

    - This is assuming that the ML-based algorithm always retrieves the ranges of the training data set. The fact that the ranges were within the realistic roam of aerosol profiles and properties is not a weakness, but a goal for a robust inversion ML algorithm.

*(21) page 11, Fig. 5: When you wrote earlier that the mean error is "-0.14", you really just took the mean over all angles? What is the significance of this? If it were in have the parameter space +50% and in the other -50%, its mean is still 0... and the model really not good, so what is the significance of the mean error here?*

    - We agree, that as a stand-alone mean error over all observation geometries and all aerosol scenarios is somewhat meaningless unless the goal is to detect any systematic biases. That is why we also show 2 standard deviation results and dependency on observation geometry.

*(22) page 12, Fig. 7: please explain the box whisker plot. Is the line mean or median? The box is how much percentage? The whisker? There are different conventions...*

    - we added the following text to the Figure 5 caption: "The central mark indicates the median, and the bottom and top edges of the box indicate the 25th and 75th percentiles, respectively. The whiskers extend to the most extreme data points not considered outliers, and the outliers are plotted individually using the '+' symbol."

*(23) page16, line351: This paper does not present the ML-based algorithm. It presents some of the results and that's it. There is not enough information on the ML model. This sentence is not summarizing the paper.*

    - we have made modifications to expend the discussion of the ML-based algorithm

*(24) page 16, line 363: maybe this is because of the choice of training parameters as a linear distributed AOD?*

    - while the AOD itself is linearly distributed the dAMF used in training is not. The more realistic reason for the lower accuracy for low AOD is probably a smaller signal in dAMF.

**Technical corrections and suggestions:**

*(1) Many times, there are definite articles missing (e.g. page 3 line 65 "The MAXDOAS..", page 3 line 84: "The DOAS technique", page 4 line 91 "The offset term...")*

    - thank you for pointing this out

*(2) Eq. 5 on page 4 is not referred to in the text.*

    - we added a reference to (Eq. 5) on line 90.

*(3) page7, line204: I highly doubt that Clemer et al 2010 is the only code here. I would add a "e.g.".*

    - added

*(4) page 9, line 229: I think you miss the AOD=0 case in this list*

    - we assumed that the algorithm will retrieve the properties perfectly in the absence of noise and AOD = 0 (dAMF = 0) and did not want to skew the data. However, we will include very small AOD in the next version of the model with the real data.

*(5) page 10ff: I suggest to use an equi-distant grid for the raa-sza plots, as they are now, it gives a biased impression to the eye.*
 - we replaced Fig 4.

*(6) page 12, line 311 f: I cannot quite understand the sentence "This error contribution..." maybe you can reformulate*
 - the phrase was replaced with "Layer partial AOD retrieval error relative to the total AOD"

**References**

P. Tan, M. Steinbach, A. Karpatne, and V. Kumar "Introduction to Data Mining (2nd Ed.)," Pearson Addison–Wesley, ISBN-13: 978-0133128901, 2018.

Friedman, Jerome, Trevor Hastie, and Robert Tibshirani. *The elements of statistical learning*. Vol. 1, no. 10. New York: Springer series in statistics, 2001.

**Responses to the comments by Reviewer 2.**

*1. Not enough information about the machine learning (ML) algorithm itself are given. The introduction focuses on aerosols and MAX-DOAS only without introducing ML. In section 5, there is no explanation why the individual ML steps were chosen as they are.*
 - We have removed some details about the aerosols and have added the following section to the introduction: See response to comment 1 by Reviewer 1.

*2. The validation section appears to be insufficient to assess the performance of the algorithm:*
*(a) Why not changing the testing dataset to realistic profiles which are not included in the training data? How can you be sure that you do not over-fit your results?*

 -The mappings contained in the test set are different from those in the training set. And different combination of SSA/ASY/profile produces different values of radiance. The model hasn't seen these input values and output combinations of SSA/ASY/profile before. We split the data into two sets, the training set and the test set, no automatic model selection process using validation set. The training loss converges. We use 75% of the entire dataset for training and then directly apply the model on the remaining 25% for testing. We use ReLU unit (through sparsity) and Max Pooling layer (through reducing parameters) to control overfitting.

*(b) Why not using larger aerosol loads?*

 - Lower to medium AOD loadings are more common. We will use LIVAS data base in future studies with more realistic profiles and global AOD loadings

*(c) Why did you use 16 different elevation angles for the testing dataset even though this number is much too high for most measurement locations? What happens if you just use 8 or 10 elevation angles? Does the algorithm still perform well?*

410

- The main goal of the current work is to evaluate feasibility of ML as a retrieval method. We have included more angles than typical for MAX-DOAS to explore the maximum information content of the measurements and the inversion method. We have also tried a scan with10 angles and the ML methods performed well.

415 *(d) The training dataset was created by using an US standard atmosphere. This is mostly a poor representation of the true atmosphere. What happens if the conditions change?*

We have not evaluated the effect of temperature and pressure profiles on the
420 retrieval at this stage. This will be done in the future studies

*(e) Why not testing the algorithm on real data?*
We do not have real data at 360 nm with accurate partial AOD profiles, ASY and SSA retrieved in the same direction as MAX-DOAS pointing.

425

*3. Are there plans to extend the training dataset by more wavelengths, SSA/asymmetry factors, albedo, profiles, trace gases (as suggested in Sec. 7)?*
- Yes

430 *4. Note that many articles are missing in the manuscript.*
- we have gone through the references
*Specific comments*
*P2, L36-42: There is no need to show the equation of SSA and its detailed description. I suggest to change those 6 lines into one sentence only.*
435 - we have removed details about SSA from the manuscript

*P2, L43-62: This part about the aerosol phase functions is much too long, especially since there is no further discussion about this topic in your paper. The lines about the Legendre expansion could be completely removed without loosing important*
440 *information for the understanding of the manuscript.*
- we have removed details about ASY from the manuscript

*P2, L58: When kept, please change the index L of PL(cosθ) P3, L65: "The" MAX-DOAS*
445 - we have removed details about ASY from the manuscript

*Figure 1: A single sky scan...*
- corrected

450 *P3, L84: "The" DOAS technique...*
- corrected

*P4, L91: "The" offset term...*
         - corrected

455

*P4, L101: "Forward model parameters that are considered approximately". I guess*
*you are referring, among other parameters, to a priori knowledge when using*
*"approximately" here? Please change the wording or reformulate this sentence.*
         - this is reference to such parameters as temperature and pressure profiles.
460       Clarified in the text now: (e.g., temperature and pressure profiles from
         atmospheric soundings or models).

*P4, L114-115: "A priori information about...". I find this sentence to be rather*
*confusing. What do you want to say here?*
465       - we added two commas to improve readability: "…distribution, before the
         measurements are made…"

*P5, L123-125: "None of the algorithms perform perfectly". That depends on what you*
*understand as "perfect". I don't think that it is possible at all to retrieve the true*
470   *atmosphere in a extremely high vertical resolution. However, as far as I know, the*
*second part of this sentence is correct. I would suggest to reformulate the first part.*
         - We removed this sentence.
*Note that you also used external information about the atmosphere for creating your*
*training dataset. You directly applied a priori knowledge by using exponentially*
475   *decreasing profiles including Gaussian's for your dataset. And I would not say that a*
*priori knowledge does not exist. You look out of the window and know that it is a hazy*
*day so you adapt the a priori. Sometimes you know about local sources or have*
*ancillary measurements available. I would strongly suggest to change this paragraph.*
         - a priori information, as applied in MAP, is typically a climatological distribution
480       of the parameters of interest so adjusting the a priori according to the observation
         at the moment is not an appropriate use of the technique. The only technique that
         is available to measure aerosol extinction coefficient vertical profiles at 355 nm is
         LIDAR but even in this case the aerosol property information is very limited, so
         we do not agree that the true a priori for AOD, ASY and SSA at 360 nm exists for
485       many locations. However, since there might be some locations where some of this
         information is available we have replaced the sentence with:
          "They also require external information about the atmosphere (e.g. temperature
         and pressure profiles) that might not be readily available at the measurement time
         scales, and a priori information might not exist'
490

*P7, L176: AMF represents → An AMF represents*
         - changed

*P7, L178: observations → observation*
495       - changed

**P7, L181: Where "the" vertical column density**

- changed

500     *P7, L183-191: This part could be shortened as you have already introduced aerosols before.*
        - we removed some of the introductory aerosol information.

        *P7, L205: "The" VLIDORT code*
505     - changed

        *Table 1: Why do your Gaussian profiles don't have center heights higher than 2km? Since the vertical sensitivity for higher altitudes is an issue for common algorithms, I am wondering if your algorithm performs better here?*
510     - in our opinion the limiting factor for the retrieval of elevated layers is the MAX-DOAS approach not the specific algorithm itself. By subtracting the zenith AMF we are removing some information. This information is also reduced by typically considering only a "single species" ($O_4$) at a "single wavelength". Because of this it made sense to limit the tests to 2 km. Further studies will include actual
515     measurements compiled in LIVAS database.

        *What is the scaling height of your exponential functions?*
        - they recalculated depending on the total loading and partitioning between the Gaussian and exponential AOD.
520
        *P9, L235: What was the reason for changing the grid step width to a coarser resolution for higher altitudes? I have the feeling that your choice of Gaussian profile center heights and retrieval grid steps might deteriorate retrieval results for higher altitudes (as indicated in Fig. 9 and 11).*
525     - Yes, this is correct. Since this is a feasibility study we first wanted to demonstrate that the method works before performing much more elaborate RT and retrieval modeling. We plan to expand the study including higher vertical resolution.

530     *Section 5: I think it would be nice to add more information to this section to explain also the in-between steps and parameters of your CNN and LSTM.*
        - we have added more details about the ML in the text and supplement

        *P9, L251: "RMSprop was chosen...". Please explain.*
535     - It is a consensus in the CS community that RMSprop works well on recurrent networks such as LSTM, but RMSprop is an unpublished optimization algorithm.

        *Figure 4: It would be interesting for the reader to see a similar plot describing the profile shape distribution. You could show 3 more plots for different partitioning,*
540     *showing Gaussian center heights and width on x and y axis, respectively. Furthermore, the number of profiles with a certain total AOD would also be interesting (especially when looking at Figure 7). I fear that the reader might loose the connection to the actual profile shape due to the rather statistical analysis in the following paragraphs.*

545

- while this is an interesting information we feel it does not add any additional insight into the results;

*Figure 5, 6 and 8: It is interesting to see that mean error and standard deviation show areas with high or low values at certain geometries. I was wondering if this is a matter of the scattering angle (angle between incident and outgoing photon assuming single*
550 *scattering)? Could you please create a plot showing the scattering angle versus the respective*
*error/standard deviation? Since e.g. RAA = 30◦ and a low SZA is equivalent to a large scattering angle (e.g. Fig 5c) it might show issues for certain scattering geometries. In addition, you could check if there are certain profile shapes or aerosol parameters*
555 *more frequent for areas with a large standard deviation or high mean errors compared to other geometries. I was also wondering about the outliers in all three histograms. Any reason for that?*

- Thank you for the recommendation! Since Henyey-Greenstein approximation has a poor representation of the forward and backward scattering we will apply
560 the suggested analysis to the future more realistic aerosol modeling.

*P11, L292: I agree that OEM methods also struggle with data inversion measured at small RAA but I was wondering why your synthetic analysis fails?*

- We believe this is due to the RT at small RAA, where the photon paths are very
565 "direct" and MAX-DOAS is not "benefitting" from the low elevation angles as much.

*P12, L295: "The total AOD retrieval..." or "The retrieval of total..."*

- changed
570

*P12, L297: In general, "the" ML algorithm*

- changed

*P12, L299: What is the reason for the second peak in the histogram?*
575 *- we do not know*

*Figure 7: Please explain all depicted quantities (mean, median, percentiles...) in the caption of this figure.*

- added: "The central mark indicates the median, the bottom and top edges of the
580 box indicate the 25th and 75th percentiles, respectively. The whiskers extend to the most extreme data points not considered outliers, and the outliers are plotted individually using the '+' symbol."

*Here, it would also be interesting to see if the largest underestimations correspond to*
585 *certain profiles or parameters.*

- figure 11 shows some of the worst cases that correspond to low AOD, RAA <= $10^o$ and large SZA >= $80^o$.

*Figure 9: Why is the error larger for 1.5 km than for 2 km? Since you also included Gaussian's with Peak heights around 2km, I would expected the largest error at higher altitudes. Especially when considering the higher sensitivity of MAX-DOAS measurements for aerosol loads closer to the surface (which can be seen for altitudes lower than 1.5km).*

      - This is potentially an artifact of the layer depths changes at 1 and 3 km

*Figure 10: It appears that there is also an underestimation of the predicted AOD for all sub-figures with true partial AOD's larger than 0.2. For example in the upper left subfigure, but also in the second row (first figure), the third row (2nd and 3rd fig). Do these underestimations correspond to problematic scenarios/geometries/parameters?*

      - We have not explored the details.

*P16, L353: Training and evaluation of "the" ML*

      - changed

*P16-17, L365-372: Points 1 and 2 are valid but only a demand for near real-time applications. I doubt that there is a need for science to have profiles immediately after the measurement. For point 3, I was wondering if this is a major advantage. The dependence of profiling results on SSA is rather small and the Henyey-Greenstein approximation is in most cases a poor representation of the scattering distribution of aerosols. So why should the reader decide for your algorithm when an AERONET station nearby measures "real" phase functions and SSA?*

      - this is a feasibility study and by no means suggests that the presented algorithm should be used as is. However, what this study does suggest is that more elaborate RT modeling with more realistic RT settings and realistic profiles can open the possibilities to fast and potentially more accurate algorithms using multiple wavelengths and multiple species.

*In point 4 you even diminish the potential of your approach by saying that it might be used as an initial guess for other algorithms. To me, this does not sound as if the authors are convinced of the capability of ML algorithms.*

      - The goal of this research is not to convert the entire community to use ML-based approach but rather to explore its feasibility and possibilities

*If this is true, why not? If not, why are there no strong arguments in favor of your approach?*

      - We personally believe that physics based ML methods can be very effective in accurate inversions, especially of MAX-DOAS data. However, the quality of training data is very important. Ideally, 3D models should be used with complete physics and exhaustive atmospheric conditions. Availability of such dataset is the part that we are mostly skeptical about. Another issue is the validation of the actual retrievals. There are no other profilers that "sample" in a similar way.

635 *track changes version:*

**A feasibility study to use machine learning as an inversion algorithm for aerosol profile and property retrieval from multi-axis differential absorption spectroscopy measurements.**

640 Yun Dong[1], Elena Spinei[1], Anuj Karpatne[2]

[1]Department of Electrical and Computer Engineering, Virginia Tech, Blacksburg, VA 24060, USA
[2]Department of Computer Science, Virginia Tech, Blacksburg, VA 24060, USA

*Correspondence to*: Elena Spinei (eslind@vt.edu)

**Abstract.** In this study, we explore a new approach based on machine learning (ML) for deriving aerosol
645 extinction coefficient profiles, single scattering albedo and asymmetry parameter at 360 nm from a single
MAX-DOAS sky scan. Our method relies on a multi-output sequence-to-sequence model combining
Convolutional Neural Networks (CNN) for feature extraction and Long Short-Term Memory networks
(LSTM) for profile prediction. The model was trained and evaluated using data simulated by VLIDORT v2.7,
which contains 1459200 unique mappings. 75% randomly selected simulations were used for training and
650 the remaining 25% for validation. The overall error of estimated aerosol properties for (1) total AOD is -1.4
± 10.1 %, (2) for single scattering albedo is 0.1 ± 3.6 %; and (3) asymmetry factor is -0.1 ± 2.1 %. The
resulting model is capable of retrieving aerosol extinction coefficient profiles with degrading accuracy as a
function of height. The uncertainty due to the randomness in ML training is also discussed.

**1. Introduction**

655 Aerosols play an important role in the Earth-atmosphere system by modifying the global energy balance,
participating in cloud formation and atmospheric chemistry, and fertilizing land and ocean. Aerosols are
widely spread in the troposphere and are emitted by anthropogenic and natural processes (primary aerosols),
and are formed by gas-to-particle conversion mechanisms (secondary aerosols). Aerosols are removed from
the atmosphere by dry (gravitational settling and turbulent) deposition and wet deposition, and have variable
660 lifetimes ranging from a few minutes to a few weeks (Haywood and Boucher, 2000).

The spatial and temporal distribution of aerosols in the lower troposphere is highly variable and greatly
depends on the proximity to the sources, type of aerosols, meteorological conditions, and photochemical
processes. Horizontal and vertical heterogeneity of the aerosol distribution, their properties and processes
pose a serious challenge for modeling aerosol induced radiative forcing and is an important source of
665 uncertainties in the climate modeling results (Intergovernmental Panel on Climate Change, 2014).

| Deleted: relatively short |
| --- |
| **Deleted:** The aerosol classification depends on the aerosol source, composition, size and number distribution, aging processes, and optical and physical properties. |
| **Deleted:** is |
| **Deleted:** affected by |

Macroscopic aerosol optical properties required for modeling aerosol radiative forcing include single scattering albedo, scattering phase function, and aerosol optical thickness (AOD), (Dubovik et al., 2002).

This paper investigates the potential of using advances in machine learning to invert aerosol properties (aerosol extinction coefficient profiles, single scattering albedo and scattering phase function) from a hyperspectral remote sensing technique called multi-axis differential optical absorption.

**Machine learning (ML)** is a branch of artificial intelligence that derives its roots from pattern recognition and statistics. The goal of ML is to build statistical (or mathematical) models of a real-world phenomenon by relying on training examples. For instance, in supervised ML, a model is first presented with a set of paired examples (termed as the training set), where every training example contains a pair of input variables and output variables, and the goal of ML algorithms is to find the statistical structure of mapping from the input variables to the output variables that match with the training examples and can be generalized to unseen examples (termed as test set). The learned mapping (or the model) can be applied to the inputs of test examples to make predictions on their outputs. There are several advantages of using ML. Firstly, it can sift through vast amounts of training data and discover patterns that are not apparent to humans. Secondly, ML algorithms can have continuous improvement in accuracy and efficiency with increasing amount of training data. Thirdly, ML algorithms are usually very fast to apply on test examples since the time-consuming training process of ML models is offline and one-time. With these advantages as well as the availability of faster hardware, ML has soon become the most popular data analytic technique since the 1990s. In recent years, it has also been applied to the field of remote sensing (Efremenko et al., 2017; Hedelt et al., 2019).

**Artificial neural networks** (ANN) are methods studied in the ML field, successfully applied to a number of commercial problems such as image detection, text translation, and speech recognition. It is inspired by the biological neural networks constituting animal brains. As an analogy to a biological brain, an ANN is based on artificial neurons. An artificial neuron is a mathematical function receiving and processing input signals and producing outputs signals or activations. Each neuron comprises of weighted inputs, an activation function, and an output. Weights of the neuron are parameters to be adjusted, while the activation function defines the relationship from the input signals to the output signals. When multiple neurons are composed together in a layered manner (where the output signals of neurons in a given layer are used as inputs for the neurons in the next layer), we call it an artificial neural network (ANN). A common algorithm for training ANNs is the backpropagation algorithm, that passes the gradients of errors on the training set from the output layer to inner layers to refine the weights at all layers in an incremental way. The backpropagation algorithm converges when there is no change in ANN weights across all layers beyond a certain threshold. There are several optimization methods that are used for performing backpropagation and are behind standalone ANN packages commonly used by the ML community. ANNs

715 have many different types depending on the specifics of the neuron arrangement or architecture. A simple type of ANN is a multilayer perceptron (MLP), where all neurons at a given layer are fully connected with all neurons of the next layer, also termed as dense layers. Other complex types of ANN include convolutional neural network (CNN) and recurrent neural network (RNN). Two important types of artificial neural networks used in this study are the convolutional neural networks (CNN) (Fukushima,

720 1980; LeCun et al., 1999) and the Long short-term memory (LSTM) neural networks (Hochreiter and Schmidhuber, 1997), which are variants of recurrent neural networks.

**Convolutional neural network (CNN)** is a class of deep neural networks that uses the convolution operation to define the type of connections from one layer to another. While they have shown impressive results in extracting complex features from images in computer vision applications (Krizhevsky et al.,

725 2012; Simonyan and Zisserman, 2015), they are relevant in many other applications involving structured input data, e.g., 1D-sequences. A CNN is composed of an input layer, multiple hidden layers and an output layer. The hidden layers usually consist of several convolutional layers, followed by pooling layers, fully connected layers (dense layers) and normalization layers. Figure 1 shows a simple example of CNN. The input vector (or sequence) is first passed through a convolutional layer where it is convolved with 3 filters

730 (convolution kernels) of size 3 using the same padding to produce three 6x1 feature maps. Since the ReLU function $f(x) = max(0, x)$ is commonly chosen as the activation function in CNNs, the feature maps only contain positive values. Then the max pooling layer picks the maximum value every 3 elements for each feature map, generating three 2 x 1 vectors. After passing through a flatten layer, the max pooling output is reshaped into a 6 x 1 vector, which is followed by a dense (fully connected) layer with 2 nodes.

735 The dense layer multiplies its input by a weight matrix and add a bias vector for generating the output of the model. The computer adjusts the model's convolutional kernel values or weights through a training process called backpropagation, a class of algorithms utilizing the gradient of loss function to update weights. For the case in Figure 1, there are 26 tunable parameters. $((3 + 1) \times 3 = 12$ from convolution kernels and $(6 + 1) \times 2 = 14$ from the dense layer.)

[Figure]

740

Figure 1. Schematics of a simple CNN

**Long short-term memory (LSTM) neural networks** have many applications such as speech recognition (Li and Wu, 2015) and handwriting recognition (Graves et al., 2008; Graves and Schmidhuber, 2009). They are a special kind of ANNs termed as recurrent neural networks (RNNs). RNNs are designed for modeling sequence dependent behavior (e.g., in time). They are called "recurrent" because they perform the same operation for every element of a sequence, with the output at a given element dependent on previous computations at earlier elements (Britz, 2015). This is different from traditional neural networks wherein all the input-output examples are assumed to be independent of each other.

[Figure]

**Figure 2. Unrolled recurrent neural network.**

Figure 2 shows a diagram of an unrolled RNN with $t$ input nodes, where "unrolled" means showing the network for the full sequence of inputs and outputs. The RNNs work as follows. At the first element of the sequence, the set of input signals $x_1$ (which can be multi-dimensional) is fed into the neural network F to produce an output $h_1$. At the next element of the sequence, the same neural network F takes both the next input $x_2$ and previous output $h_1$, generating the next output $h_2$. This recurrent computation continues for t times to produce the output at the $t^{th}$ element of the sequence, $h_t$. While RNNs are powerful architectures for modeling sequence behavior, classical RNNs are inadequate to capture long-term memory effects where the inputs-outputs at a given element of the sequence can affect the outputs at another element of the sequence separated by a long interval. Long-short-term memory (LSTM) models are variants of RNNs that are able to overcome this challenge and are efficient at capturing long-term dependencies as well as short-term dependencies. It does so by introducing an internal memory state that is operated by neural network layers termed as gates, such as the "input gate," that adds new information from the input signals to the memory state, the "forgot gate," that erases content from the memory state depending on the input signals, and the "output gate," that transforms information contained in the input signals and the memory state to produce output signals.

**Figure 3.  LSTM cell diagram (modified from Thomas, 2018).**

770    An example of an LSTM cell is illustrated in Figure 3, of which the update rules are:

$$g_j = \tanh\left(b^g + x_j U^g + h_{j-1} V^g\right)$$

$$i_j = \sigma(b^i + x_j U^i + h_{j-1} V^i)$$

$$f_j = \sigma(b^f + x_j U^f + h_{j-1} V^f)$$

$$s_j = s_{j-1} \circ f_j + g_j \circ i_j$$

775

$$o_j = \sigma(b^o + x_j U^o + h_{j-1} V^o)$$

$$h_j = \tanh(s_j) \circ o_j$$

where $j$ is the element index, $\sigma(x)$ represents the sigmoid function, and $\tanh(x)$ represents the hyperbolic tangent function. $x \circ y$ denotes the element-wise product of $x$ and $y$. $U^g, U^i, U^f, U^o$ are the weights for the input $x_j$, while $V^g, V^i, V^f, V^o$ are the weights for the other input $h_{j-1}$, and $b^g, b^i, b^f, b^o$ are the scalar terms

780    (termed as bias). The term $g_j$ is the input modulation gate, which modulates the input $b^g + x_j U^g + h_{j-1} V^g$ by a hyperbolic tangent function, squashing the input between -1 to 1. The term $i_j$ is the input gate, which applies a sigmoid function to its input, limiting the output values between 0 and 1. The input gate $i_j$ determines which inputs are switched on or off when multiplying the modulated inputs $(g_j \circ i_j)$. The term $s_j$ is the internal cell state that provides an internal recurrence loop to learn the sequence dependence. The terms

785    $f_j$ and $o_j$ are the forgot gate and output gate, respectively. They have similar function to the input gate $i_j$, regulating the information into and out of the LSTM cell. The term $h_j$ is the output at step $j$.

**2. Multi-Axis Differential Optical Absorption (MAX-DOAS) technique**

[revised manuscript text omitted]

Scalar calculations;

Only elastic scattering;

Aerosol scattering phase function estimation using Henyey-Greenstein approximation from the asymmetry factor (g). | **Observation Geometry:**
Viewing zenith angle scan: 0, 40, 50, 60, 65, 70, 75, 80, 81, 82, 83, 84, 85, 86, 87, 88, 89°;
Relative azimuth angles: 0, 10, 20, 30, 40, 50, 60, 70, 80, 90, 100, 110, 120, 130, 140, 150, 160, 170, 180°
Solar Zenith angles: 0, 10, 20, 30, 40, 50, 60, 65, 70, 75, 80, 85, 86, 87, 88, 89°

**Wavelength:** 360 nm;

**Vertical grid (67 layers):**
100 m up to 4 km, 500 m from 4 to 8 km, 1 km from 8 to 12km, 2 km from 12 to 30km, 5 km from 30 to 60 km

**Atmospheric air density:**
Pressure [hPa]: US1976 standard atmosphere
Temperature [K]: US1976 standard atmosphere

**Gas volume mixing ratio profiles:**
$O_3$ profile: climatology over Cabauw in September
$O_3$ molecular absorption cross-section: Daumont
$O_2O_2$ profile: from temperature and pressure
$O_2O_2$ molecular absorption cross-section: Thalman and Volkamer (2011)

**Aerosol properties:**
Single scattering albedo: 0.775, 0.825, 0.875, 0.925, 0.975
Henyey-Greenstein asymmetry factor: 0.675, 0.725, 0.775, 0.825

**Aerosol extinction coefficient profiles [1/km] as a function of altitude;**
Exponential function at the surface combined with "sliding" Gaussian function above;
Total AOD: 0, 0.15, 0.3, 0.45, 0.6, 0.75;
Gaussian profile center height: 0.5, 1, 1.5, 2 km;
Gaussian width: 0.1, 0.2, 0.3, 0.5 km;
Partitioning between exponential and Gaussian attributed AOD: 0.3, 0.6, 0.9

**Surface reflectivity:**
Lambertian albedo at 0.04 |

VLIDORT models radiative transfer processes at a specific wavelength in a stratified atmosphere. It requires geometrical and "optical" information about the atmospheric layers and the underlying ground surface. These

975 include layer heights, pressure and temperature at layer boundaries for refractive geometry calculations, solar zenith, viewing zenith direction and relative azimuth angles between the viewing direction and solar position. Each atmospheric layer is described by total optical thickness, total single scatter albedo, and the set of Greek matrices specifying the total scattering law.

VLIDORT simulations were performed for the US 1976 standard atmosphere divided into 67 layers (same

980 as in Frieß et al., 2019) with 0.1 km layers from the surface to 4 km; 0.5 km layers from 4 to 8 km and varying width up to 60 km. Since surface reflectivity has a small effect on ground-based MAX-DOAS measurements

we performed simulations only for a single Lambertian albedo of 0.04. Absorption only by two gases was considered in this study: ozone and $O_2O_2$. Light polarization, direct beam refraction, and inelastic scattering were not included in this study. Table 1 summarizes VLIDORT inputs and general settings.

Aerosol types in this study are described by a single scattering albedo and asymmetry factor combination with total 20 "types": (1) Single scattering albedo: 0.775, 0.825, 0.875, 0.925, 0.975; (2) Henyey-Greenstein asymmetry factor: 0.675, 0.725, 0.775, 0.825. Aerosol extinction coefficient profiles were generated by combining an exponential function at the surface with a "sliding" Gaussian function above. The aerosol total optical depth was partitioned between the exponential and Gaussian functions. Total AOD cases included 0.15, 0.3, 0.45, 0.6, and 0.75 with exponential to Gaussian partitioning fractions of 0.3, 0.6 and 0.9. The Gaussian function peak center height was varied from 0.5 km to 2 km in steps of 0.5 km. The Gaussian function peak width was varied too: 0.1, 0.2, 0.3, and 0.5 km. This results in 4800 aerosol cases and a total of 1459200 measurement simulations (sky scan). Figure 14 demonstrates the aerosol profile samples, where the near surface aerosol partial optical depth profiles are described by the exponential function and the layers aloft are described by the Gaussian function with various widths and heights added to the exponential function profile. While VLIDORT simulations were performed for an atmosphere divided into 67 layers, ML training was done by resampling onto 23 layers only. The new layer depths are: 100 m from the surface to 1km, 200 m from 1 km to 3 km, 500 m from 3 km to 4 km, and the last layer is 56 km high. The new layer partial AODs were generated by adding the neighboring layer partial aerosol optical depths. ML algorithm was trained on 75% randomly selected measurement simulations (1094400 samples) and model performance was tested on the remaining 25%. Note, that no validation data was held off from the 75% training set for tuning hyper-parameters of our ML model, as all ML hyper-parameters were kept constant across all experimental settings in this paper.

**5. Learning inverse mapping using ML**

We employ a supervised ML formulation for our problem of aerosol profile retrieval, where the goal is to learn the mapping from input variables to output variables given a training set of paired data instances. In our formulation, every data instance corresponds to a single MAX-DOAS sky scan at a fixed Relative Azimuth Angle (RAA) and Solar Zenith Angle (SZA), where the inputs of the data instance comprise of: (a) RAA scalar value, (b) SZA scalar value, and (c) a sequence of $\Delta AMF^{aerosol}$ values at 16 VZAs. The output variables at a data instance correspond to the aerosol properties we are interested in predicting given the inputs, which are: (a) Single Scattering Albedo (SSA) scalar value, (b) Asymmetry factor (ASY) scalar value, and (c) a sequence of partial Aerosol Optical Depth (AOD) values at 23 vertical layers of the atmosphere, termed as the aerosol extinction profile.

Note, that in our supervised ML formulation, there are sequences in both the input signals and output signals, namely $\Delta AMF^{aerosol}$ sequence and partial AOD sequence, respectively. Further note that the input and output signals used in our problem setting are of very different types and thus have different dimensionalities (e.g.,

$\Delta AMF^{aerosol}$ takes 16 values at varying VZAs while partial AOD takes 23 values at varying atmospheric layers). We thus first apply a 1-dimensional CNN to extract features from the sequence part of the input signals. Note that our input signals are not image-based, which is one of the common types of input data for which CNNs are used. Instead, our input data is structured as a 1D sequence, and the convolution operations of CNN help in extracting sequence-based features from the input signals that are then fed into subsequent ANN components. We also use an LSTM to model the sequence part of the output signals. Note, that our data contains no time dimension as we are only working with single scan data, assuming the atmosphere does not change during the scan time. However, it is the sequence-based nature of the output signals that motivated us to use LSTM models for sequence-based output prediction. Furthermore, the dataset we use for training is produced by a physical model (VLIDORT), where the relationship between the inputs and outputs are known.

Figure 6 illustrates the novel multi-output sequence-to-sequence model for learning the inverse mapping from MAX-DOAS measurements to aerosol optical properties. To extract sequence-based features from MAX-DOAS inputs, a 1-dimensional Convolutional Neural Network (CNN, Fukushima, 1980; LeCun et al., 1999) is first applied on the sequence of inputs (we concatenate $\Delta AMF^{aerosol}$ sequence with SZA and RAA to obtain an 18-length input sequence), which results in a sequence of preliminary hidden features. These preliminary hidden features are then sent to two different branches of 1D-CNN layers that perform further compositions of convolution operators to produce non-linear hidden features for predicting two different types of outputs: (a) scalar outputs: SSA and ASY, and (b) sequence-based outputs: aerosol extinction profile. For the branch corresponding to scalar outputs, the features extracted from 1D-CNN layers are simply passed on to a fully-connected dense layer to produce a two-dimensional output of SSA and ASY. For the branch corresponding to sequence-based outputs, the features extracted from 1D-CNN layers are fed to a Long Short-Term Memory network (LSTM, Hochreiter and Schmidhuber, 1997) to produce a sequence of partial AOD values at varying atmospheric layers.

[Figure]

**Figure 6. Schematics of the multi-output sequence-to-sequence model for deriving aerosol optical properties from MAX-DOAS measurements.**

Figure S1 shows the detailed architecture of the multi-output sequence-to-sequence model. The CNNs consist of eight 1D convolutional layers ($c_1$ to $c_8$) and four max-pooling layers ($p_1$ to $p_4$). For convolutional layers $c_1$ to $c_6$, the activation function is the Rectified Linear Unit (ReLU) function. For layers $c_7$ and $c_8$, it

is a hyperbolic tangent function (tanh). We set the kernel size of the convolution operation to be the typical value of 5 and use the same padding for all $c_k, \forall k \in \{1, 2, ... ,8\}$. ReLU and Max pooling layers help to reduce overfitting through model sparsity and parameter reduction. The convolution kernel weights are initialized using a "Glorot uniform" method (Glorot  and Bengio, 2010).

Extracted feature vector from the $p_1$ layer is sent into two different branches. In the branch for profile prediction, we take a one-to-many LSTM (Fig. 3) with 23 layer steps and a hidden size of 128 to capture the correlation between the partial AODs at different layers. We simply duplicate the feature vector learned from CNNs for 23 times to generate the inputs for the LSTM model. The sequential output $\{y_1, y_2,….,y_{23}\}$ of the LSTM (after passing through a flatten layer and an ReLU layer) is interpreted as the 23-layer aerosol extinction profile. For the SSA/ASY branch, 1D convolutional layers and dense layers are combined for the prediction. The reason for taking a two-output architecture is that SSA and ASY are independent scalar outputs that cannot be treated as a sequence, in contrast to the aerosol extinction profile.

We implemented our ML model in the Jupyter Notebook using the Keras library, which is a commonly used deep learning library for Python. RMSprop was chosen as the optimizer and the mean squared error was used as the loss function (Hinton, 2012). We trained the model on 75% of the dataset for 124 epochs with a batch size of 640. The following choice of hyperparameters was used: choice of optimizer = RMSprop, lr = 0.001, rho = 0.9, epsilon = None, and decay = 0.0. We did not perform any hyper-parameter tuning on a separately held validation set inside the training set, and the values of all hyper-parameters in our ML model were kept constant throughout all experiments in the paper on the test set. In order to ensure that there was no overlap between the training and testing steps, we did not make use of the test data either directly or indirectly during the training phase, either for learning parameter weights or selecting hyper-parameters.

**6. Results**

Evaluation of the accuracy of ML mapping rules derived during the training stage for MAX-DOAS data inversion was done by comparing the "true" atmospheric aerosol properties to the ML inverted properties. The evaluation data set consists of 364800 MAX-DOAS simulated sky scans that are outside of the training set. The number of simulations in the evaluation data set as a function of solar zenith angle (SZA) and relative azimuth angle (RAA) are shown in Figure 7. Between 1100 and 1300 aerosol scenarios are present in each SZA-RAA bin.

[Figure]

[Figure]

1130   **Figure 7. Number of simulations in the evaluation data set as a function of solar zenith (SZA) angle and relative azimuth angle (RAA).**

The following ML predicted aerosol properties were evaluated: (1) asymmetry factor, (2) single scattering albedo, (3) total aerosol optical thickness, and (4) partial aerosol optical thickness for each layer from 0 to 4 km. A relative error $\epsilon$ of the retrieved by ML parameter $\hat{x}$ relative to the "true" value $x$ is calculated according

1135   to Eq. (10):

$$\epsilon \equiv \frac{\hat{x} - x}{x} \cdot 100\% \,, \tag{6}$$

The relative error evaluation presented in the subsequent sections was performed on the retrievals from a single ML training. Since ML itself introduces randomness during the training stage, we retrained the model 20 times with the same hyperparameters for evaluating the uncertainty of the ML training.

1140   **6.1. Asymmetry factor at 360 nm**

The ML-based approach shows an ability to invert aerosol asymmetry factor with a mean error of -0.14% and two standard deviations of 2.04% and nearly normal error distribution (Fig. 8(a)). To evaluate if any dependence of the asymmetry factor retrieval exists on SZA and RAA the mean error and the two standard deviations are shown in Fig. 8(b, c). These distributions suggest that dependence of the asymmetry factor

1145   retrieval on SZAs and RAAs is relatively small. However, systematically higher relative errors are observed around SZA of 65° and RAA of 30-40°.  The cause of these elevated errors is not clear at this point.

[Figure]

**Figure 8. Asymmetry factor retrieval errors: (a) error histogram; (b) mean error as a function of SZA and RAA; (c) two standard deviations as a function of SZA and RAA.**

1150   **6.2. Single scattering albedo at 360 nm**

Similar high accuracy is achieved for ML retrieval of the single scattering albedo with a mean error of 0.19% and two standard deviations of 3.46% and nearly normal error distribution, somewhat positively skewed (Fig. 9). Slightly higher errors are observed at RAA smaller than 60° and most SZA.

[Figure]

1160 **Figure 9. Single scattering albedo retrieval errors: (a) error histogram (b) mean error as a function of SZA and RAA (c) two standard deviations as a function of SZA and RAA.**

Mean errors are also larger at small RAA and SZA > 85°. Traditional optimal estimation techniques also struggle with the MAX-DOAS data inversion at small RAA due to uncertainty in aerosol forward and backward scattering.

1165 **6.3. Total aerosol optical depth at 360 nm**

Total AOD retrieval is more challenging for the ML model than the single scattering albedo or asymmetry factor, especially at lower total AOD levels. Box plots of the total AOD error for different "true" total AOD values are given in Fig. 10. In general, ML algorithm tends to underestimate total AOD from the mean error ± 2 standard deviations of -8.39 ± 8.81% (total AOD 0.15) to -1.52 ± 3.10% (total AOD of 0.75). Total

1170 AOD retrieval error distribution over all cases is close to Gaussian distribution, but with two peaks (Fig. 11). The mean error (± two standard deviations) is -3.58% ± 7.68%. The bias of the model does not have much dependence on SZAs and RAAs (Fig. 11(b)). Still, lager errors and uncertainties can be observed at higher SZAs and lower RAAs (Fig. 11(c)).

[Figure]

**Figure 10. Box plots of total AOD prediction errors for each "true" total AOD value. The box central mark indicates the median, and the bottom and top edges of the box indicate the 25th and 75th percentiles, respectively. The whiskers extend to the most extreme data points not considered outliers, and the outliers are plotted individually using the '+' symbol.**

[revised manuscript text omitted]

Britz, D.: Recurrent Neural Networks Tutorial, Part 1 – Introduction to RNNs, WildML [online] Available from: http://www.wildml.com/2015/09/recurrent-neural-networks-tutorial-part-1-introduction-to-rnns/

1300  (Accessed 15 January 2020), 2015.

Clémer, K., Van Roozendael, M., Fayt, C., Hendrick, F., Hermans, C., Pinardi, G., Spurr, R., Wang, P. and De Mazière, M.: Multiple wavelength retrieval of tropospheric aerosol optical properties from MAXDOAS measurements in Beijing, Atmospheric Measurement Techniques, 3(4), 863–878, doi:10.5194/amt-3-863-2010, 2010.

1305  Dong, Y., Spinei, E. and Karpatne, A.: amt-2019-368, University Libraries, Virginia Tech [online] Available from https://doi.org/10.7294/6A3T-ZV25, 2019.

Dubovik, O., Holben, B., Eck, T. F., Smirnov, A., Kaufman, Y. J., King, M. D., Tanré, D. and Slutsker, I.: Variability of Absorption and Optical Properties of Key Aerosol Types Observed in Worldwide Locations, Journal of the Atmospheric Sciences, 59(3), 590–608, doi:10.1175/1520-

1310  0469(2002)059<0590:VOAAOP>2.0.CO;2, 2002.

Efremenko, D. S., Loyola R., D. G., Hedelt, P. and Spurr, R. J. D.: Volcanic SO2 plume height retrieval from UV sensors using a full-physics inverse learning machine algorithm, International Journal of Remote Sensing, 38(sup1), 1–27, doi:10.1080/01431161.2017.1348644, 2017.

Frieß, U., Beirle, S., Alvarado Bonilla, L., Bösch, T., Friedrich, M. M., Hendrick, F., Piters, A., Richter, A.,

1315  van Roozendael, M., Rozanov, V. V., Spinei, E., Tirpitz, J.-L., Vlemmix, T., Wagner, T. and Wang, Y.: Intercomparison of MAX-DOAS vertical profile retrieval algorithms: studies using synthetic data, Atmospheric Measurement Techniques, 12(4), 2155–2181, doi:10.5194/amt-12-2155-2019, 2019.

Fukushima, K.: Neocognitron: A self-organizing neural network model for a mechanism of pattern recognition unaffected by shift in position, Biol. Cybernetics, 36(4), 193–202, doi:10.1007/BF00344251,

1320  1980.

Glorot, X. and Bengio, Y.: Understanding the difficulty of training deep feedforward neural networks, 8, 2010.

Graves, A. and Schmidhuber, J.: Offline Handwriting Recognition with Multidimensional Recurrent Neural Networks, in Advances in Neural Information Processing Systems 21, edited by D. Koller, D. Schuurmans,

1325  Y. Bengio, and L. Bottou, pp. 545–552, Curran Associates, Inc. [online] Available from: http://papers.nips.cc/paper/3449-offline-handwriting-recognition-with-multidimensional-recurrent-neural-networks.pdf (Accessed 4 January 2020), 2009.

Graves, A., Liwicki, M., Bunke, H., Schmidhuber, J. and Fernández, S.: Unconstrained On-line Handwriting Recognition with Recurrent Neural Networks, in Advances in Neural Information Processing Systems 20, edited by J. C. Platt, D. Koller, Y. Singer, and S. T. Roweis, pp. 577–584, Curran Associates, Inc. [online] Available from: http://papers.nips.cc/paper/3213-unconstrained-on-line-handwriting-recognition-with-recurrent-neural-networks.pdf (Accessed 4 January 2020), 2008.

Haywood, J. and Boucher, O.: Estimates of the direct and indirect radiative forcing due to tropospheric aerosols: A review, Reviews of Geophysics, 38(4), 513–543, doi:10.1029/1999RG000078, 2000.

Hedelt, P., Efremenko, D. S., Loyola, D. G., Spurr, R. and Clarisse, L.: SO2 Layer Height retrieval from Sentinel-5 Precursor/TROPOMI using FP_ILM, Atmospheric Measurement Techniques Discussions, 1–23, doi:10.5194/amt-2019-13, 2019.

Hinton, G.: Neural Networks for Machine Learning Lecture 6a, [online] Available from: https://www.cs.toronto.edu/~tijmen/csc321/slides/lecture_slides_lec6.pdf (Accessed 16 March 2019), 2012.

Hochreiter, S. and Schmidhuber, J.: Long Short-Term Memory, Neural Computation, 9(8), 1735–1780, 1997.

Intergovernmental Panel on Climate Change, Ed.: Evaluation of Climate Models, in Climate Change 2013 - The Physical Science Basis, pp. 741–866, Cambridge University Press, Cambridge., 2014.

Johansson, E. m., Dowla, F. u. and Goodman, D. m.: Backpropagation learning for multilayer feed-forward neural networks using the conjugate gradient method, Int. J. Neur. Syst., 02(04), 291–301, doi:10.1142/S0129065791000261, 1991.

Krizhevsky, A., Sutskever, I. and Hinton, G. E.: ImageNet Classification with Deep Convolutional Neural Networks, in Advances in Neural Information Processing Systems 25, edited by F. Pereira, C. J. C. Burges, L. Bottou, and K. Q. Weinberger, pp. 1097–1105, Curran Associates, Inc. [online] Available from: http://papers.nips.cc/paper/4824-imagenet-classification-with-deep-convolutional-neural-networks.pdf (Accessed 4 January 2020), 2012

LeCun, Y., Haffner, P., Bottou, L. and Bengio, Y.: Object Recognition with Gradient-Based Learning, in Shape, Contour and Grouping in Computer Vision, edited by D. A. Forsyth, J. L. Mundy, V. di Gesú, and R. Cipolla, pp. 319–345, Springer Berlin Heidelberg, Berlin, Heidelberg., 1999.

Li, X. and Wu, X.: Constructing Long Short-Term Memory based Deep Recurrent Neural Networks for Large Vocabulary Speech Recognition, arXiv:1410.4281 [cs] [online] Available from: http://arxiv.org/abs/1410.4281 (Accessed 16 January 2020), 2015.

Platt, U. and Stutz, J.: Differential optical absorption spectroscopy: principles and applications, Springer, Berlin., 2008.

Rodgers, C. D.: Inverse methods for atmospheric sounding: theory and practice, Reprinted., World Scientific, Singapore., 2004.

Schilling, H., Bulatov, D., Niessner, R., Middelmann, W. and Soergel, U.: Detection of Vehicles in Multisensor Data via Multibranch Convolutional Neural Networks, IEEE Journal of Selected Topics in Applied Earth Observations and Remote Sensing, 11, 4299–4316, doi:10.1109/JSTARS.2018.2825099, 2018.

Schulz, K., Hänsch, R. and Sörgel, U.: Machine learning methods for remote sensing applications: an overview, in Earth Resources and Environmental Remote Sensing/GIS Applications IX, vol. 10790, p. 1079002, International Society for Optics and Photonics., 2018.

Simonyan, K. and Zisserman, A.: Very Deep Convolutional Networks for Large-Scale Image Recognition, arXiv:1409.1556 [cs] [online] Available from: http://arxiv.org/abs/1409.1556 (Accessed 4 January 2020), 2015.

Spurr, R.: LIDORT and VLIDORT: Linearized pseudo-spherical scalar and vector discrete ordinate radiative transfer models for use in remote sensing retrieval problems, in Light Scattering Reviews 3, edited by A. A. Kokhanovsky, pp. 229–275, Springer Berlin Heidelberg, Berlin, Heidelberg., 2008.

Thalman, R. and Volkamer, R.: Temperature dependent absorption cross-sections of O2–O2 collision pairs between 340 and 630 nm and at atmospherically relevant pressure, Physical Chemistry Chemical Physics, 15(37), 15371, doi:10.1039/c3cp50968k, 2013.

Thomas, A.: Keras LSTM tutorial - How to easily build a powerful deep learning language model, Adventures in Machine Learning [online] Available from: https://adventuresinmachinelearning.com/keras-lstm-tutorial/ (Accessed 16 March 2019), 2018.

[revised manuscript text omitted]